# Turbulence Distortion Matters in Predicting Inflow Turbulence Noise of Future Wind Turbines

Özgür Yalçın[1], Andrea Piccolo[1], Riccardo Zamponi[1,2], Daniele Ragni[1], and Roberto Merino-Martinez[3]

[1]Department of Flow Physics and Technology, Delft University of Technology, Kluyverweg 1, 2629HS Delft, the Netherlands
[2]Environmental and Applied Fluid Dynamics Department, von Karman Institute for Fluid Dynamics, Waterloosesteenweg 72, B-1640 Sint-Genesius-Rode, Belgium
[3]Department of Control and Operations, Delft University of Technology, Kluyverweg 1, 2629HS Delft, the Netherlands

**Correspondence:** Özgür Yalçın (o.e.yalcin@tudelft.nl)

**Abstract.** This study examines the role of turbulence distortion in predicting inflow turbulence (IT) noise generation from large wind turbines via Amiet's theory. Two subsequent distortion mechanisms are investigated: (i) the streamtube expansion in the rotor induction zone and (ii) the interaction with the surface of thick blade profiles. Large-eddy simulations reveal that the turbulence spectra, which reflect distortion effects, remain largely unaffected by rotor induction within the frequency range relevant for noise generation. As for the other mechanism, the distortion of the turbulence approaching a blade leading edge is modeled with a simplified closed-form solution of Goldstein's Rapid Distortion Theory. This vorticity-deflection-based model is extended here beyond the high-frequency approximation and integrated into an analytical Amiet-based IT noise tool. Applications to representative test cases show that while distortion effects are minimal for current turbine sizes, they become relevant for future configurations featuring larger rotor sizes and thicker airfoils. The developed model reveals that IT noise levels do not necessarily scale with rotor size, but are shaped by spectral changes induced by the blade geometry, operational parameters, and inflow conditions. This model offers a physically consistent, computationally efficient framework for aeroacoustic assessment of next-generation wind turbine design.

### Nomenclature

#### Coordinates and frames

$(x_1, y_1, z_1)$  translated reference frame

$(x_2, y_2, z_2)$  rotated reference frame

$(x_a, y_a, z_a)$  Amiet-oriented reference frame

$(x_h, y_h, z_h)$  hub-based reference frame

$\hat{s}$  unit direction vector from the retarded source to the observer

$\sigma$  amplified distance

| | | |
|---|---|---|
| $\boldsymbol{x}$ | | observer position in Amiet frame |
| $\boldsymbol{x}_o$ | | observer position in hub frame |
| $\boldsymbol{x}_s$ | | present source position in hub frame |
| $\xi$ | | dummy spatial coordinate |
| $r$ | | radial location of a segment |
| $r_e$ | | distance between the observer and the retarded source position |
| $r_o$ | | direct observer distance from the hub |
| $R_y$ | | rotation matrix about $y$-axis |
| $R_z$ | | rotation matrix about $z$-axis |

**Flow Kinematics and Mach Numbers**

| | | |
|---|---|---|
| $\Omega$ | | blade angular velocity |
| $\rho_\infty$ | | freestream density |
| $c_\infty$ | | speed of sound |
| $M$ | | incoming Mach number along $x_a$ |
| $M_b$ | | rotational blade Mach number |
| $M_c$ | | convection Mach number |
| $M_s$ | | azimuthal Mach number |
| $M_t$ | | transversal freestream Mach number |
| $M_z$ | | axial freestream Mach number |
| $M_{\mathrm{rel}}$ | | relative Mach number between the incoming flow and the moving segment |
| $U$ | | mean flow speed |
| $u$ | | turbulent velocity fluctuation |

**Angles**

| | | |
|---|---|---|
| $\alpha$ | | local angle of attack |

| | $\beta$ | local pitch angle |
| --- | --- | --- |
| 45 | | |

$\beta$    local pitch angle

$\beta_0$    blade pitch angle

$\gamma$    local azimuth angle

$\psi$    angle of $M_t$ relative to $y_h$-axis

$\Theta$    angle between $M_c$ and $\boldsymbol{x}_o$

$\theta$    angle of $r_o$ relative to $z_h$-axis

**Rotor and Geometry**

AR    area ratio

$a_x$    axial induction factor

$B$    number of blades

$b$    span of a segment

$c$    chord of a segment

$D$    rotor diameter

$N_\mathrm{seg}$    number of segments per blade

$R$    cylinder/rotor radius

**Turbulence and Aeroacoustics**

$\mathcal{L}$    aeroacoustic transfer function

$\omega$    Doppler-shifted frequency

$\omega_e$    emission frequency

$\phi_{11}$    streamwise energy spectrum

$\phi_{22}$    upwash energy spectrum

SPL    sound pressure level

TI    turbulence intensity

$f$    frequency

$k$        wavenumber

$L$        incoming turbulent length scale

$L_w$      sound power level

$l_y$       correlation length

$S_{22}$     power spectral density of upwash velocity fluctuations

$S'_{pp}$     radiated power spectral density at an azimuthal angle

**Rapid Distortion Theory**

$(\infty)$       denotes undistorted variable

$(X_1, X_2, X_3)$  drift coordinates

$\boldsymbol{\kappa}$        distorted wavenumber domain

$\boldsymbol{u}^{(g)}$      gust velocity

$\boldsymbol{v}$        unsteady velocity

$\hat{}$        denotes complex amplitude

$\mathrm{D}/\mathrm{D}t$    substantial derivative

$\sigma_s$       distance from the upstream along the streamline

$\varphi$        velocity potential term

$a$        magnitude of the gust velocity

$D_{ij}$       distortion tensor

$l_{\mathrm{dist}}$      distortion length scale

$Q$        magnitude of the velocity potential

# 1 Introduction

The recent increased demand for renewable energy has promoted the deployment of new, larger wind turbines (WTs) (Global Wind Energy Council, April 2024; Wind Europe, February 2025). One of the major concerns for onshore WTs, however, remains their noise emissions (Kirkegaard et al., 2025), which restrict their placement close to residential areas due to national noise regulations (Davy et al., 2018). In modern WTs, mechanical noise has been significantly reduced thanks to advances in gearbox technology, making aerodynamic noise the predominant source (Rogers and Manwell, 2004; Oerlemans et al., 2007). Aerodynamic noise can be classified mainly into blade self-noise and inflow turbulence (IT) noise. Considering blade optimization and operational flow regimes, trailing edge (TE) noise is a dominant self-noise source, which inherently scatters in the mid-to-high frequency range (Oerlemans et al., 2007). On the other side, IT noise arises from the interaction between the incoming turbulent flow and the blade leading edges (LEs), predominantly emitting low-frequency noise ($f < 200\,\mathrm{Hz}$, corresponding to acoustic wavelengths larger than blade chords and comparable to large WT rotor scales) (Lowson, 1993; Buck et al., 2016).

Recent developments, such as TE add-ons and active flow control mechanisms, have yielded notable reductions in TE noise (Zhou et al., 2022; Lahoz et al., 2024). However, increasing rotor sizes and hub heights of next-generation turbines shifts the noise spectrum to lower frequencies, thus amplifying the relevance of IT noise (Møller and Pedersen, 2011). This shift is particularly important considering that low-frequency sound waves propagate over longer distances and experience reduced air absorption and transmission losses by building materials. Therefore, they are more perceptible indoors due to room resonances, which increase their negative health impact (Blumendeller et al., 2020). Consequently, accurate prediction of IT noise has become as critical as TE noise for the development of quieter WTs.

Several low-fidelity approaches for predicting IT and TE noise in rotational motion (Moriarty and Migliore, 2003; Sinayoko et al., 2013; Tian et al., 2013; Buck et al., 2016; Botero-Bolívar et al., 2024) have been developed as extensions of Amiet's theory (Amiet, 1975), which was initially proposed for rectilinear motion. These methods represent an efficient approach in terms of cost and accuracy, particularly for IT noise predictions, since the relevant flow region just upstream of the blade LE is minimally affected by the boundary layer development. Indeed, Amiet's model, originally formulated for a flat plate with a relatively large span immersed in a subsonic turbulent flow, relates the far-field acoustic power spectral density to the incoming gust velocity spectrum via a transfer function. This means that IT characteristics are the only required inputs to obtain an estimate of the acoustic performances.

A primary limitation of Amiet's model lies, therefore, in its dependence on the IT spectrum, which is inherently difficult to predict and measure. In practice, the inflow is assumed to be isotropic and characterized by canonical spectra, such as von Kármán or Liepmann models, parameterized by upstream turbulence intensity and integral length scale. However, experimental and numerical/analytical studies have demonstrated that the upstream turbulence distorts as it approaches the LEs of thick airfoil sections (Batchelor and Proudman, 1954; Lighthill, 1956; Hunt, 1973; Paterson and Amiet, 1976), resulting in an alteration of its velocity spectrum. The key mechanism has been identified as the distortion of the vorticity field in the stagnation region, which reduces the upwash velocity in the wavenumber range associated with turbulent scales smaller than the characteristic

length (e.g., LE radii (Mish and Devenport, 2006) or maximum thickness (Gill et al., 2013)). This distortion leads to a marked increase in the slope of the upwash velocity spectrum at high frequencies, resulting in a corresponding attenuation of noise
emissions in the same range (Atassi et al., 1990; Moreau and Roger, 2005; Mish and Devenport, 2006; Christophe, 2011). On the one hand, this highlights the influence of airfoil thickness on sound generation; on the other, it demonstrates that neglecting this effect can lead to discrepancies between predicted and measured IT noise levels (Devenport et al., 2010; Santana et al., 2016; Buck et al., 2018; Zamponi et al., 2020; Piccolo et al., 2023). In particular, it is important to use the velocity spectrum sampled or modeled as close to the LE as possible, as input to Amiet's model.

To address this limitation, several enhancements have been proposed to incorporate turbulence distortion and thickness effects (Moreau and Roger, 2005; Santana et al., 2016; dos Santos et al., 2023, 2024), also for WTs, considering their relatively thick blade profiles (Moriarty et al., 2005; Buck et al., 2018; Faria et al., 2020; Botero-Bolívar et al., 2024). These modifications typically adjust empirical parameters within the canonical turbulence spectrum, particularly targeting its inertial subrange to better capture variations in spectral slope and hence improve IT noise predictions. Some studies have also explored these
adaptations for rotational motion, such as using one-dimensional turbulence spectra to integrate single-hot-wire measurements (Piccolo et al., 2025).

While these corrected models have shown better agreement with experimental data in wind tunnel flows over moderately thick airfoils, their applicability to full-scale WT operations, where incoming turbulence has integral length scales much larger than the airfoil chord, is questionable (Faria et al., 2020; Botero-Bolívar et al., 2024). These models may underpredict IT noise
under atmospheric turbulence conditions, where canonical models sometimes perform better (Botero-Bolívar et al., 2024).

Alternatively, the Rapid Distortion Theory (RDT), that is based on linearized vorticity equations, can represent a valid approach to account for the distortion effects, as it allows for a more generalized and realistic framework. In principle, it models the distortion of weak turbulent perturbations under mean flow convection conditions in cases where the coherence between eddies is unaltered. Originally developed by Batchelor and Proudman (1954) and extended by Hunt (1973), the RDT
formulates distortion through a set of coupled partial differential equations to describe the upstream field of a two-dimensional bluff body by wavenumber analysis. It distinguishes two mechanisms according to the relative size of the eddies with respect to a characteristic geometric dimension of the body. The first is the blockage due to the body surface, and the second is the vorticity deflection due to the deformation of the mean flow streamlines.

Although the asymptotic results of the RDT have already been incorporated into Amiet's model as corrections in the men-
150 tioned studies, Zamponi et al. (2021) proposed a way to implement the full solution of Hunt's formulation. The resulting estimation of distortion effects could support the development of a tool for assessing the performance of noise-reduction technologies. However, the considerable complexity and high computational cost of the implementation restricted the wavenumber range over which solutions could be obtained, limiting the practicality of this methodology for low-fidelity applications.

An alternative RDT approach, originally proposed by Goldstein (1978), is based on a linearized Euler equation for non-
155 uniform, homentropic mean flows. Solving these equations using a Lagrangian framework in drift coordinates allows for direct tracking of turbulent flow in the distorted field. This method is especially suitable for low-speed flows, such as those typical for

the upstream regions of the WT blade profiles. For high-frequency gusts, a simplified version of this model was later derived by Majumdar and Peake (1998) to estimate fan inflow noise.

Turbulence distortion is not only related to the thickness of the aerodynamic object in the convective direction. In wind energy applications, the expansion of the streamtube induced by the WT rotor also plays a role by creating a region of flow deceleration. Unlike contracting flows in wind tunnels or propulsion systems, streamtube expansion is associated with non-uniform and inhomogeneous strain-rate fields in the induction zone, violating key assumptions of considering distortion as rapid (Batchelor and Proudman, 1954). As a result, the distortion of large energetic eddies, associated with the low-frequency components, which are typically of the order of a rotor diameter, is not adequately captured by classical RDT approaches (Mann et al., 2018; Ghate et al., 2018; Milne and Graham, 2019). Other theoretical studies based on linearized or quasi-steady fluctuation models, which show qualitative agreement with high-fidelity simulations and measurements, found that low-frequency eddies undergo significant distortion, whereas high-frequency ones are largely unaffected (Graham, 2017; Mann et al., 2018; Ghate et al., 2018; Milne and Graham, 2019). In terms of noise emissions, it should be noted that the sound generated by distorted large-scale turbulent structures in the inflow lies in the very low-frequency range (below $\sim 1\,\mathrm{Hz}$), and is therefore outside the audible range (Buck et al., 2018). IT noise for WTs, usually prevailing in the $1\text{-}200\,\mathrm{Hz}$ range of sound waves, is instead generated by the interaction of small-scale turbulent structures, i.e., those associated with mid-to-high turbulence frequencies, with blades. Yet, it remains unclear whether turbulence within the IT noise frequency band is exposed to noticeable distortion due to streamtube expansion and, if so, how to best model it. This highlights a key gap in current understanding.

This study examines the impact of turbulence distortion on IT noise in next-generation large-scale onshore WTs, considering the thickening blade profiles (Veers et al., 2019; Timmer and Bak, 2013; Schaffarczyk et al., 2024; Global, 2024) and intensifying inflow conditions that cannot be readily reproduced in wind tunnel experiments (Saab et al., 2018). This analysis is carried out using a developed Amiet's model framework. The distortion of turbulence is considered as two subsequent mechanisms. First, the high-frequency distortion due to streamtube expansion is evaluated using atmospheric flow simulations performed with the SOWFA (Simulator fOr Wind Farm Applications) (Fleming et al., 2013). Then, the distortion due to thick blade profiles is computed through a simplified closed-form solution of Goldstein's RDT associated only with vorticity deflections, allowing Amiet's model to maintain its low computational cost while enhancing its accuracy. The paper is structured as follows: after an overview of the Amiet-based IT noise tool and its adaptation to rotating frames, the turbulence distortion mechanisms and their implementations are presented. Finally, using the developed prediction model, the relevance of IT noise for future onshore WTs is analyzed, taking into account the effects of turbulence distortion.

## 2  IT noise prediction tool

The IT noise prediction tool used in this study is based on Amiet's model (Amiet, 1975), an analytical theory developed to estimate the far-field noise spectrum radiated from an infinite-span flat plate subjected to a subsonic turbulent gust. In the context of IT noise, the pressure jump across the flat plate that is responsible for acoustic radiation is considered to arise from the interaction of the LE with the incoming turbulence. The original configuration simply corresponds to an isolated airfoil

in an acoustic wind tunnel where the observer and the airfoil are fixed relative to a reference frame, as illustrated in Fig. 1. To extend this model to WT applications, the developed tool adapts Amiet's formulation to a WT rotating blade frame, by following well-established procedures for helicopters and rotors (Amiet, 1989; Sinayoko et al., 2013). This adaptation ensures that the fundamental assumptions of the model are still satisfied while accounting for the unsteady kinematics and spatial inhomogeneity present in actual WT operations.

## 2.1 Adaptation of Amiet's model to a WT reference frame

Consider an observer positioned far away from a WT, as shown in Fig. 2(a), with the reference frame initially located at the rotor hub. The adaptation procedure for this configuration consists of the following steps: (i) blade segmentation, (ii) reference frame transformation, (iii) Doppler shift effect, (iv) Application of Amiet's model, and (v) azimuthal averaging. The details of each step are provided in the following subsections.

### 2.1.1 Blade segmentation

Each WT blade is discretized into multiple segments along the span to capture the variations in inflow turbulence and blade geometry (e.g., chord, twist, and airfoil profile). Each segment is modeled as an independent IT noise source under rectilinear motion, following the original Amiet's configuration (Fig. 1). A representative segment is highlighted in orange in Fig. 2(a).

### 2.1.2 Reference frame transformation

In order to apply Amiet's formulation, the observer distance and the incoming Mach number are expressed in the local frame of each segment. This new reference frame, $(x_a, y_a, z_a)$, is centered at the midspan of a blade segment and oriented according to Amiet's coordinate system (see Fig. 2(a)). Following the procedures described by Amiet (1989) and Sinayoko et al. (2013), the required coordinate transformation is carried out in three steps:

- The hub-based reference frame, $(x_h, y_h, z_h)$, is translated to the LE of the segment's midspan, resulting in $(x_1, y_1, z_1)$;

– A rotation is applied about the $z_1$-axis by the local azimuth angle, $\gamma$, aligning the translated frame with the segment span, as shown in Fig. 2(b);

- The resulting frame, $(x_2, y_2, z_2)$, is rotated about the $y_2$-axis by an angle $\beta$, which is the local pitch angle (sum of the blade pitch and local twist angles) to ensure chord alignment, as shown in Fig. 2(c).

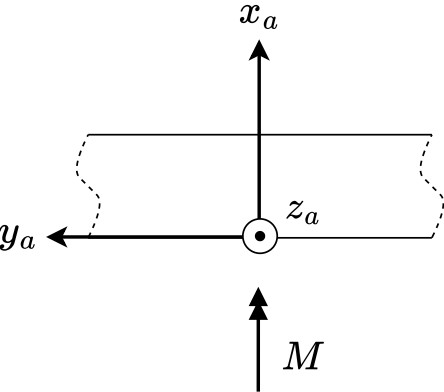

**Figure 1.** Original Amiet's coordinate system.

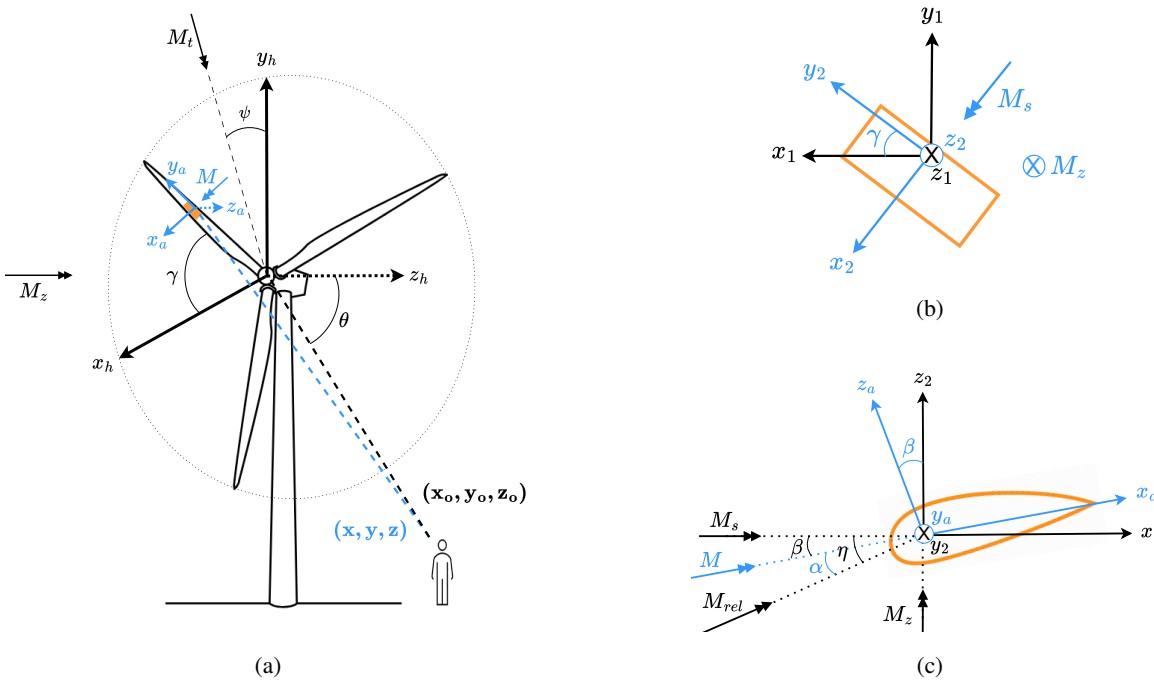

(a)

(b)

(c)

**Figure 2.** Sketches depicting coordinate transformation for adapting Amiet's model to WT rotating frames: (a) for coordinate transformation from hub to Amiet's frame for a blade segment while (b) and (c) for coordinate rotations about $z_1$- and $y_2$-axes, respectively.

This transformation yields the observer location, $\boldsymbol{x}$, relative to the local Amiet-aligned source frame, $(x_a, y_a, z_a)$. The final coordinates are calculated using the rotation matrices as follows:

$$\boldsymbol{x} = R_y(\beta) R_z(\pi/2 - \gamma)(\boldsymbol{x}_o - \boldsymbol{x}_s)$$

$$= \begin{bmatrix} \cos(\beta) & 0 & \sin(\beta) \\ 0 & 1 & 0 \\ -\sin(\beta) & 0 & \cos(\beta) \end{bmatrix} \begin{bmatrix} \cos(\pi/2 - \gamma) & -\sin(\pi/2 - \gamma) & 0 \\ \sin(\pi/2 - \gamma) & \cos(\pi/2 - \gamma) & 0 \\ 0 & 0 & 1 \end{bmatrix} (\boldsymbol{x}_o - \boldsymbol{x}_s),$$

(1)

where $\boldsymbol{x}_o$ and $\boldsymbol{x}_s$ denote the observer and the present segment/source positions based on the hub, respectively. Using the angles for the same segment demonstrated in Fig. 2, the incoming Mach number, $M$, aligned with the $x$-axis as in the original configuration is obtained by

$$M = M_{\text{rel}} \cos(\alpha).$$

(2)

Here, $M_{\text{rel}}$ is the relative Mach number between the incoming flow and the moving segment, and $\alpha$ is the local angle of attack, obtained as

$$M_{\text{rel}} = \sqrt{M_s^2 + M_z^2}$$

$$= \sqrt{(M_b + M_t \cos(\psi + \gamma))^2 + M_z^2};$$

(3)

$$\alpha = \tan^{-1}\left(\frac{M_z}{M_s}\right) - \beta.$$

(4)

In this context, $M_z$ and $M_t$ are the axial and transversal components of the freestream Mach number, respectively, as seen in Fig. 2(a), while $\psi$ defines the orientation of $M_t$ relative to the $y_h$-axis. The azimuthal Mach number, $M_s$, is derived from the vector sum of the rotational blade Mach, $M_b = \Omega r/c_\infty$ and $M_t$, where $\Omega$ is the blade angular velocity, $r$ is the radial location of the segment, and $c_\infty$ is the speed of sound.

Lastly, the present source position, $\boldsymbol{x}_s$, at the moment the emitted sound reaches the observer (i.e., the emission moment), is calculated by convecting the emission position forward within the acoustic propagation time. The approach, valid even under non-zero angle of attack conditions, follows the method described by Sinayoko et al. (2013) as

$$\boldsymbol{x}_s = r_e\{-(M_b \sin(\gamma) + M_t \cos(\gamma) \sin(\gamma + \psi))\hat{i}$$

$$+ (M_b \cos(\gamma) - M_t \sin(\gamma) \sin(\gamma + \psi))\hat{j}\}.$$

(5)

Here, $r_e$ denotes the distance between the observer and the retarded source positions, which is computed by

$$r_e = r_o \frac{M_c \cos(\Theta) + \sqrt{1 - M_c^2 \sin^2(\Theta)}}{1 - M_c^2},$$

(6)

where $r_o$ is the direct observer distance from the hub, while $M_c$ is the convection Mach number, which is simply equal to $\sqrt{M_z^2 + M_t^2}$. The angle $\Theta$ between $M_c$ and $\boldsymbol{x}_o$ is calculated as

$$\Theta = \cos^{-1}\left(\frac{M_t \sin(\psi) \sin(\theta) + M_z \cos(\theta)}{M_c}\right),$$

(7)

with $\theta$ being the angle of $r_o$ relative to the $z_h$-axis.

### 2.1.3 Doppler shift effect

Fixing the reference frame to each segment makes the source be under rectilinear motion while the observer moves relatively, necessitating a Doppler shift correction to properly account for the circular motion in reality. Amiet's model is, thus, applied at the emission frequency, $\omega_e$, whereas the observer perceives the sound at the Doppler-shifted frequency, $\omega$. The spectral relationship at the observer position $\boldsymbol{x}$ is given by

$$S_{pp}(\boldsymbol{x},\omega,\gamma) = \frac{\omega_e}{\omega} S'_{pp}(\boldsymbol{x},\omega_e,\gamma), \tag{8}$$

where $S'_{pp}$ is the radiated power spectral density (PSD) from a segment at the azimuthal angle of $\gamma$. The Doppler shift factor, $\omega_e/\omega$, is given as

$$\frac{\omega_e}{\omega} = \frac{1 + (\boldsymbol{M}_c - \boldsymbol{M}_b) \cdot \hat{s}}{1 + \boldsymbol{M}_c \cdot \hat{s}}. \tag{9}$$

Here, the convection and blade Mach numbers are expressed in vector form as

$$\boldsymbol{M}_c = -M_t \sin(\psi)\hat{i} - M_t \cos(\psi)\hat{j} - M_z\hat{k},$$
$$\boldsymbol{M}_b = -M_b \sin(\gamma)\hat{i} + M_b \cos(\gamma)\hat{j}. \tag{10}$$

In addition, the unit direction vector from the retarded source to the observer, $\hat{s}$, is obtained as

$$\hat{s} = -\left(M_t \sin(\psi) + x_o/r_e\right)\hat{i} - \left(M_t \cos(\psi) + y_o/r_e\right)\hat{j} - \left(M_z + z_o/r_e\right)\hat{k}. \tag{11}$$

### 2.1.4 Application of Amiet's model

Amiet's model is applied to each segment by taking into account the inverse strip theory (Santana et al., 2016), which overcomes the infinite-span assumption of the model. This theory basically applies Amiet's formulation to two high-aspect-ratio segments separated by the actual span length of the corresponding segment, and then subtracts the resulting spectra. This ensures that the upwash spectrum, $\phi_{22}$, and aeroacoustic transfer (airfoil response) function, $\mathcal{L}$, are nearly independent of the spanwise wavenumber (Graham, 1970), simplifying the spectral formulation (Eq. (20) in the work of Amiet (1975)). Accordingly, for the computation of the inflow turbulence sound spectrum radiated from the midspan of each segment, the following formulation is used:

$$S'_{pp}(\boldsymbol{x},\omega_e,\gamma) = \left(\frac{\rho_\infty \omega_e z c M}{2\sigma^2}\right)^2 \frac{b}{2}|\mathcal{L}|^2 S_{22}(\omega_e)l_y(\omega_e), \tag{12}$$

where $\rho_\infty$ denotes the freestream density, and $c$ and $b$ are the blade segment's chord and span, respectively. The term $\sigma^2$ is computed as $\sigma^2 = x^2 + (1 - M^2)(y^2 + z^2)$. The transfer function $\mathcal{L}$ includes both LE scattering and TE back-scattering components, following the formulations provided by Amiet (1989) and Santana et al. (2016) (not given here for the sake of brevity). The PSD of the upwash velocity fluctuations, $S_{22}$, and the spanwise correlation length, $l_y$, are revealed with the

infinite-span assumption, making the spanwise direction homogeneous. Hence, $S_{22}$ is obtained from the 1-D turbulence energy spectrum, $\phi_{22}(k_x)$, as

$$S_{22}(\omega_e) = \frac{\phi_{22}(k_x)}{U_x}, \tag{13}$$

where $U_x$ is the incoming streamwise velocity (i.e., in the $x$-axis), whereas $k_x$ is the chordwise wavenumber that equals to $k_x = \omega_e/U_x$. The spanwise correlation length $l_y$ is computed as

$$l_y(\omega_e) = \frac{15.02 L \hat{k}_x^2}{(3 + 8\hat{k}_x^2)\sqrt{1 + \hat{k}_x^2}}, \tag{14}$$

with $\hat{k}_x = k_x/k_e$ where $k_e = 0.7468/L$, and $L$ being the incoming turbulent length scale.

In the cases of isotropic incoming turbulence (i.e., neglecting distortion effects), $\phi_{22}(k_x)$ can be modeled by a von Kármán (vK) 1-D spectrum as follows:

$$\phi_{22}(k_x) = 0.0792 \frac{\overline{u_x^2}}{k_e} \frac{3 + 8\hat{k}_x^2}{\left(1 + \hat{k}_x^2\right)^{11/6}}, \tag{15}$$

where $u_x$ is the root-mean-square of fluctuating streamwise velocity. For other cases, the calculation of the distorted spectra is given in Sect. 4.1.

### 2.1.5 Azimuthal averaging

The segment-level noise predictions are azimuthally averaged over a full blade revolution to obtain the time-averaged spectrum. This introduces an additional Doppler factor, $\omega_e/\omega$, due to the rotational retarded time effect (Amiet, 1989; Sinayoko et al., 2013) (i.e., the change of time increment from source to observer). Eventually, the total IT noise radiated by the WT is obtained by summing the contributions from all segments of blades as follows:

$$S_{pp}(\boldsymbol{x}, \omega) = \frac{B}{2\pi} \sum_1^{N_{\text{seg}}} \int_0^{2\pi} \left(\frac{\omega_e}{\omega}\right)^2 S'_{pp}(\boldsymbol{x}, \omega_e, \gamma)\, \mathrm{d}\gamma, \tag{16}$$

where $B$ is the number of blades, and $N_{\text{seg}}$ is the number of segments per blade.

## 2.2 Verification of IT noise tool

The implementation of the IT noise prediction tool described in Sect. 2.1 is verified using a benchmark case of an onshore WT for which far-field noise measurements are openly available from the zEPHYR EU-project (Christophe et al., 2022). The corresponding WT is the SIEMENS SWT-2.3-92 model, which features an $80\,\mathrm{m}$ high tower and a $92\,\mathrm{m}$ rotor diameter. As reported by Leloudas (2006), measured sound pressure levels at $100\,\mathrm{m}$ downstream of the turbine tower were converted into sound power level spectra, $L_w$, by accounting for background noise effects and other relevant measurement corrections. These spectra were provided for three operational cases varying with average wind speed $U_\infty$, rotational speed $\Omega$, and blade pitch

**Table 1.** Operational parameters of benchmark cases

| Case no | $U_\infty$ [m s$^{-1}$] | $\Omega$ [RPM] | $\beta_0$ [deg] |
|---|---|---|---|
| 1 | 6 | 13 | 3 |
| 2 | 8 | 14 | -2 |
| 3 | 9.5 | 17 | 5 |

angle $\beta_0$. The associated parameters are summarized in Table 1. The IT characteristics evaluated at hub height, including the turbulent length scale ($L =$312 m) and turbulence intensity (TI $= 10.7\%$), were obtained from atmospheric boundary layer simulations based on Weather Research and Forecasting model coupled with Large-Eddy Simulation (WRF–LES) by Kale (2024), performed as part of the zEPHYR project. In particular, these parameters were calculated for Case 2 and assumed unchanged for the other cases, which is followed in this study to ensure consistency in comparisons. In the present verification, turbulence distortion effects are not included; instead, the upwash velocity spectrum is calculated using the vK isotropic model (see Eq. (15)).

The IT noise spectra are predicted for all three operating conditions using the developed tool. Figure 3 shows a comparison between the predictions and the measured data. For additional verification, results from another independently developed Amiet-based IT noise tool (Botero-Bolívar, 2023; Botero-Bolívar et al., 2024), obtained using the same IT characteristics given above, are also included in the comparison. Although this tool differs from the current implementation in terms of vK spectrum formulation, coordinate transformation, and applied strip theory, the agreement between the two prediction tools is nearly perfect across all frequencies and cases. This close match serves as a verification of the implementation of the model. On the other hand, both prediction tools demonstrate a fair agreement with the measured data, particularly in the low-frequency range, as shown in Figs. 3(d-f), where IT noise is most pronounced. It is important to note that the TE noise is not modeled in either of the Amiet-based predictions, which explains the considerable underprediction at higher frequencies.

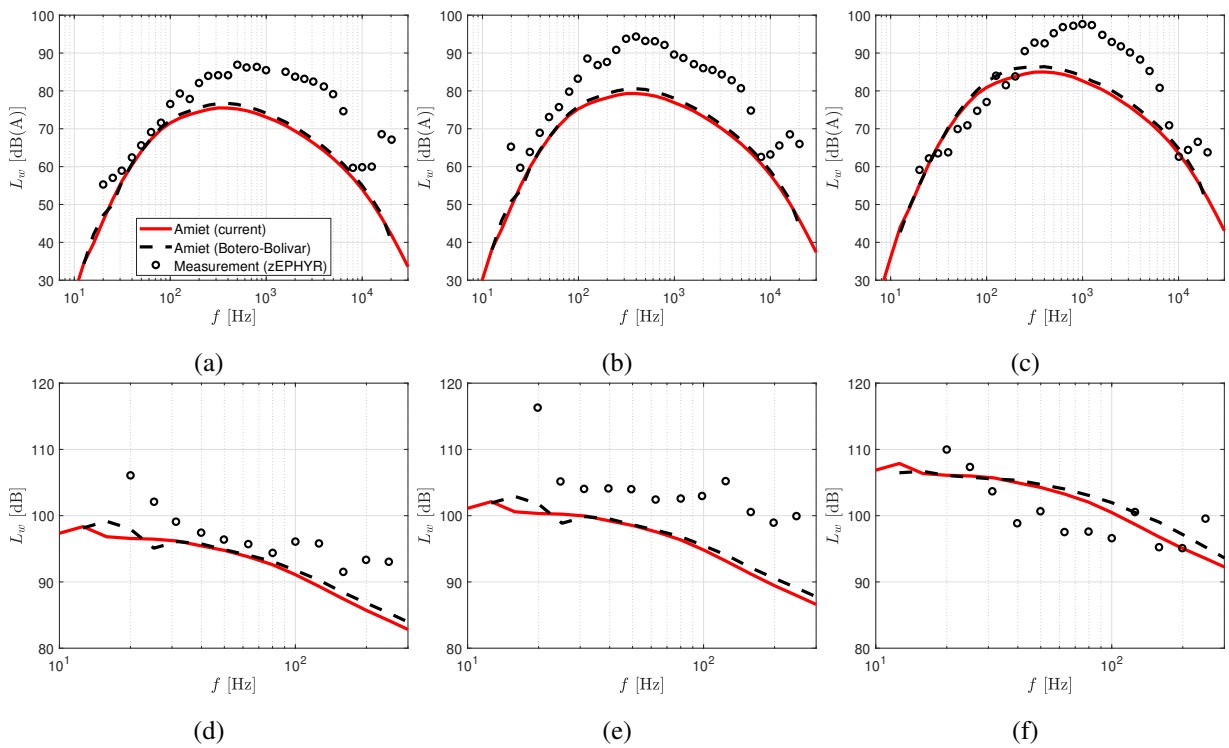

**Figure 3.** $1/3$ octave sound power levels compared with results from another Amiet tool (Botero-Bolívar, 2023) and measured data (Christophe et al., 2022). (a), (b), and (c) show A-weighting levels for Case 1, 2, and 3, respectively. (d), (e), and (f) show the levels without A-weighting for the same cases, respectively, and particularly in the low-frequency range.

## 3 Turbulence distortion due to streamtube expansion

This section investigates whether the incoming turbulence undergoes distortion during the streamtube expansion within the rotor's induction zone, with a particular focus on the frequency range relevant for IT noise. To this end, atmospheric flow simulations are performed around a large WT using the SOWFA open-source tool (Fleming et al., 2013), which basically couples the solvers OpenFOAM and OpenFAST. In this study, the flow field is simulated by Large-Eddy Simulation (LES) via OpenFOAM while modeling the rotor blade forces by Actuator Line Model via OpenFAST and projecting them to the LES grid simultaneously. The coupled framework offers a cost-effective yet sufficiently accurate solution, especially considering that accurate resolution of unsteady blade aerodynamics and wake structures is not a primary concern here. All simulations are carried out on the Delft High Performance Computing Centre (DHPC).

### 3.1 Simulation setup of SOWFA

The simulations are conducted using the NREL 5-MW reference WT, which is widely studied in the literature and whose configuration files are publicly available. Two different inflow wind speeds are considered: a below-rated case at $8\,\mathrm{m\,s^{-1}}$ and

320 an above-rated case at $14\,\mathrm{m\,s^{-1}}$. These two regimes allow for an examination of the potential influence of the axial induction factor on turbulence distortion.

Each case starts with a precursor simulation, in which the atmospheric flow is simulated without any turbine. Here, the goal is to establish a turbulent flow that has a neutral atmospheric boundary layer with a desired wind speed at the hub height of 90 m, where turbulence develops naturally through shear and surface-roughness effects. The computational domain size is 3 km 325 $\times$ 3 km $\times$ 1 km in the streamwise ($x$), spanwise ($y$), and surface-normal ($z$) directions, respectively. All horizontal boundary conditions are treated by periodicity. The streamwise and spanwise grid resolution is 10 m, whereas the grid is clustered toward the surface. The time step is chosen as 0.5 s. The precursor simulations are run for 20000 s to obtain statistically-converged turbulence.

For the below-rated case, a streamwise velocity spectrum of a random point at hub height is presented in Fig. 4. The 330 spectrum is compared with the vK spectrum, which is obtained based on the sampled turbulence intensity ($= 0.1$) and length scale ($= 150$ m), where the latter is calculated from the simulated streamwise velocity spectrum using its standard spectral definition. The results show that the flow at the hub height exhibits an isotropic turbulence behavior as the spectrum collapses onto the vK spectrum, up to a numerical cut-off frequency ($\sim 0.1$ Hz) imposed by the solver capabilities with the current grid and timestep.

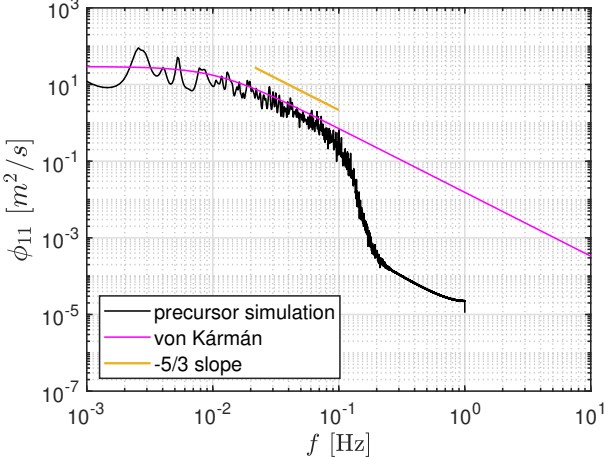

**Figure 4.** Streamwise velocity spectrum of a random point at hub height.

**3.2 Energy spectrum change in induction zone**

The last 2000 s of the flow data from the precursor simulation are stored and then used to initialize the main turbine-included simulation, serving as the initial flow domain as well as the inflow boundary condition. The grid is refined around the turbine rotor in 4 levels, with the smallest cell sizes becoming 0.625 m. Accordingly, the time step is also reduced to 0.125 s. The finest grid domain extends up to 3 rotor diameters ($D = 126$ m) upstream of the rotor plane in order to analyze turbulence distortion

in the induction zone without grid effects. Besides, the streamwise periodic boundaries are replaced by inflow and outflow conditions.

Each simulation with the modified setup and configuration is continued for $2000\,\mathrm{s}$. The turbine rotational speed is fixed at 9 RPM for both wind speed cases. During the simulations, the velocity time histories are sampled at various points in the induction zone for a period of nearly 225 revolutions of a blade at a sampling frequency of $80\,\mathrm{Hz}$. The instantaneous side and

345 cross-sectional views of streamwise velocity contours, along with the sampled data locations at hub height, are shown in Fig. 5. The turbine and its wake within the atmospheric boundary layer are clearly visible.

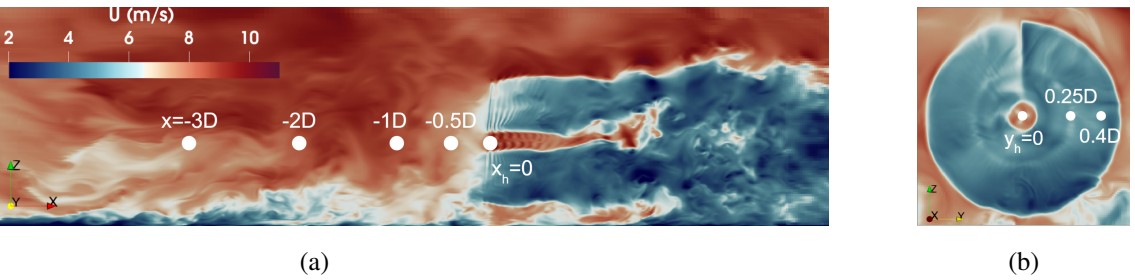

(a)    (b)

**Figure 5.** Streamwise velocity contours from different views, together with some sampled data locations: (a) for side view ($x$-$z$ plane) and (b) for cross view ($y$-$z$ plane).

The streamwise velocity component is selected for spectral analysis, as it directly corresponds to the upwash velocity with respect to the blade, which is the one responsible for inducing the unsteady surface pressure fluctuations and hence IT noise generation. The streamwise evolution of the mean and fluctuating components of this velocity in the induction zone is plotted

in Fig. 6 for both inflow cases and three different spanwise locations. Theoretical data based on the actuator disk model and the quasi-steady formulations (Mann et al., 2018) are also included for comparison. Accordingly, the theoretical distortions of the mean ($U$) and fluctuating ($u$) speed components as the flow approaches the rotor are defined as

$$\frac{U(x)}{U_\infty} = 1 - a_x \left(1 + \frac{\xi}{\sqrt{1+\xi^2}}\right)$$
$$\frac{u^2(x)}{u_\infty^2} = 1 - a_x \left(1 + \frac{\xi}{\sqrt{1+\xi^2}}\right)\left(1 + \frac{U_\infty}{a_x}\frac{da_x}{dU_\infty}\right),$$  (17)

where $a_x$ is the axial induction factor, and $\xi = 2x/D$. The induction factor and its derivative with respect to the inflow speed

are obtained from steady-state OpenFAST simulations over a range of wind speeds. Figure 6 shows that the simulated mean velocities match the theoretical predictions relatively well, while fluctuating components show only qualitative agreement, consistent with observations in earlier studies (Mann et al., 2018; Milne and Graham, 2019), which questions the validity of the linearized actuator disk theory close to the rotor. This observed decrease of the streamwise turbulence intensity in the below-rated case could be attributed to a rotor-induced blockage effect, as explained by the physical findings of the RDT.

In contrast, the above-rated case shows an amplification, with the vortex line pile-up effect likely becoming the dominant mechanism (Hunt, 1973). This difference can be related to variations in the induction factor and its gradient, both of which

modulate turbulence distortion via changes in aerodynamic loading. This is also consistent with the observed reduction in distortion towards the blade tip, where the loading is minimal.

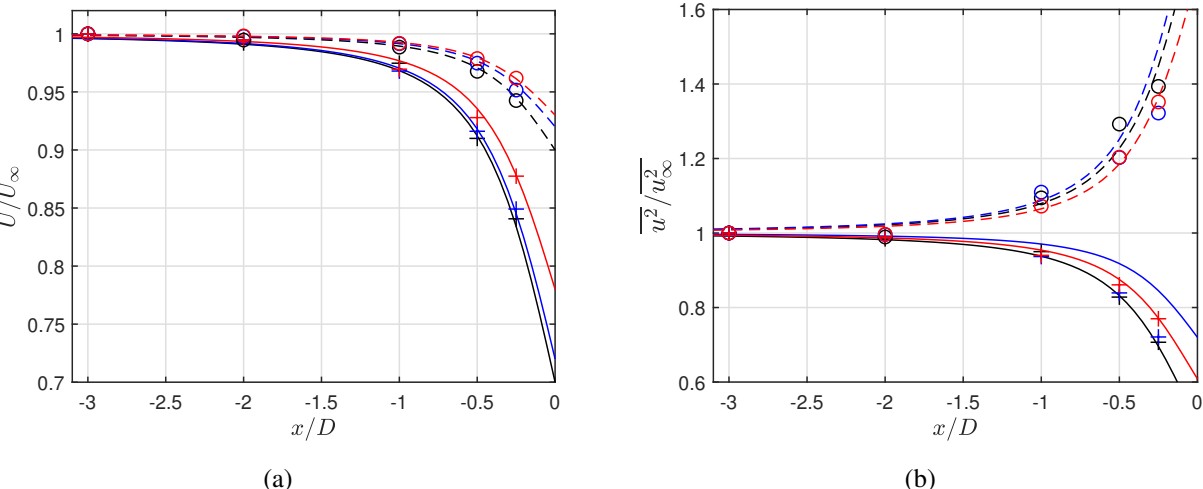

(a)        (b)

**Figure 6.** Change of mean (a) and fluctuating streamwise velocity (b) along the induction zone. Sampled data locations are shown in Figure 5. The symbols + and o denote the simulation data, whereas solid and dashed lines are generated by the theoretical formula for below- and above-rated speeds, respectively. Black, blue and red colors refer to the spanwise locations of $y_h$, $y_h + 0.25D$ and $y_h + 0.40D$, respectively.

These averaged quantities primarily reflect the dynamics of large-scale eddies, the sizes of which are larger than the rotor
diameter in both wind speed cases. Such large-scale structures correspond to very low frequencies that contribute little to audible noise radiation. Therefore, the analysis is continued by focusing on the streamwise velocity spectrum at mid-to-high frequencies, evaluated at various upstream positions for the below-rated case. A representative comparison at a spanwise location of $0.4D$, where the IT noise contribution is expected to be strong, is depicted in Fig. 7. The first observation is that the spectrum reveals the harmonics of the blade passing frequency $(0.45\,\text{Hz})$ near the rotor, superimposed on a broadband
turbulence background. Additionally, when compared to the vK model, each simulated spectrum exhibits a change in the inertial subrange slope $(-5/3)$, which becomes gradually steeper between the numerical cut-off frequency of the coarse upstream mesh $(\sim 0.1\,\text{Hz})$ and that of the refined near-rotor mesh $(\sim 1.6\,\text{Hz})$. The same behaviour appears at all upstream locations, indicating that the coarser upstream region limits the development of the corresponding small turbulent scales before they reach the rotor plane. This suggests that the observed change is numerical in origin, rather than a rotor-induced distortion. The
same conclusion is reached for the above-rated speed case (not shown here). These results must be evaluated considering the capacity of the SOWFA tool, where the near field of the rotor is modeled. Nevertheless, the findings are consistent with those of linearized-theory-based studies (Mann et al., 2018; Ghate et al., 2018; Milne and Graham, 2019), where the distortion of small-scale eddies was reported as negligible. Thus, based on these investigations, turbulence distortion due to the streamtube expansion is neglected in the remaining part of this study for the frequency range of interest.

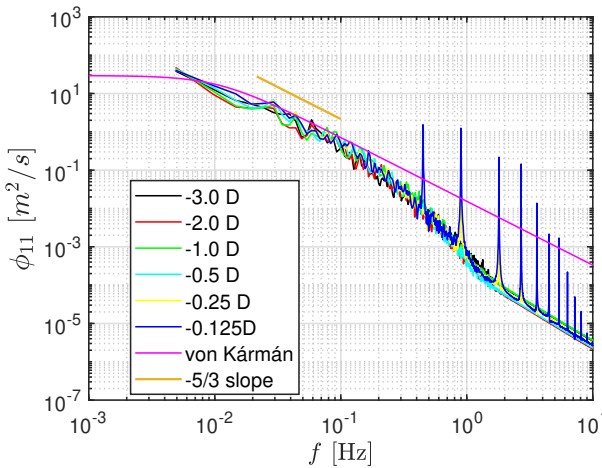

**Figure 7.** Change of streamwise energy spectrum along the upstream rotor axis ($y_h + 0.4D, z_h$).

## 4  Turbulence distortion due to blade thickness

Following the streamtube expansion, inflow turbulence is additionally affected by the blade thickness as it approaches the blade LEs. In this section, this mechanism is examined using the RDT of Goldstein (1978). A practical implementation of this approach for aeroacoustic applications has already been presented by Glegg and Devenport (2017), including a systematic method to compute distorted spectra from undistorted ones. Their formulation relies on a distortion tensor approximated for high-frequency gusts by Majumdar and Peake (1998). In this study, following the steps of Glegg and Devenport (2017), a generalized version of this approximated distortion tensor is derived, which remains valid across all frequency ranges, not limited to the high-frequency regime. The goal of this derivation is to increase the accuracy in Amiet-based IT noise predictions for WT blades with large distortion-inducing body thicknesses. The following section presents how to obtain this tensor starting from the fundamental assumptions and initial steps of Goldstein's RDT. In the end, its capability is discussed within the concept of classical distortion mechanisms.

### 4.1  Goldstein's RDT

The RDT framework assumes the following conditions:

- The incoming turbulence is weak: $u_\infty \ll U_\infty$.

- The distorted unsteady flow is isentropic and incompressible.

- The mean flow is irrotational[1] and homentropic, implying that boundary layer effects and flow separation are neglected.

---

[1]The mean flow does not have to be strictly irrotational; it is the irrotational component that primarily governs turbulence distortion.

– The distortion time scale is much shorter than the turbulent time scale. This is the main principle of RDT, meaning the disturbances are convected only by the mean flow.

Goldstein's RDT approach starts with decomposing the unsteady velocity ($\boldsymbol{v}$) as

$$\boldsymbol{v} = \boldsymbol{U} + \nabla\varphi + \boldsymbol{u}^{(g)}. \tag{18}$$

where $\boldsymbol{U}$ is the mean velocity, $\varphi$ is the velocity potential term related to pressure fluctuations, and $\boldsymbol{u}^{(g)}$ is the gust velocity including the remaining potential and rotational disturbances. At the upstream boundary, the unsteady flow is determined by the gust velocity while the velocity potential being zero. Under the above assumptions, the linearized Euler equations reduce to the wave equation for the velocity potential as

$$\frac{1}{c_\infty^2}\frac{\mathrm{D}_0^2\varphi}{\mathrm{D}t^2} - \nabla^2\varphi = \nabla\cdot\boldsymbol{u}^{(g)}, \tag{19}$$

where $\mathrm{D}_0/\mathrm{D}t$ is the substantial derivative relative to the mean flow. As the disturbances are convected by an irrotational mean flow, the upstream gust velocity can be tracked by the drift coordinates analytically, which also allows Eq. (19) to be uncoupled. Drift coordinates $(X_1, X_2, X_3)$ were initially introduced by Darwin (1953) and then Lighthill (1956) as

$$\frac{\mathrm{D}_0 X_1}{\mathrm{D}t} = U_\infty, \quad \frac{\mathrm{D}_0 X_2}{\mathrm{D}t} = 0, \quad \frac{\mathrm{D}_0 X_3}{\mathrm{D}t} = 0. \tag{20}$$

Here, the constant $X_1$ identifies the surface representing the fluid locations after the same particles are convected with the mean flow velocity for a certain time ($X_1/U_\infty$), whereas $X_2$ and $X_3$ are the stream surfaces orthogonal to each other (not to $X_1$). $X_1$ is obtained via

$$X_1 = U_\infty \int\limits_{\text{streamline}} \frac{\mathrm{d}\sigma_s}{U}, \tag{21}$$

with $\sigma_s$ denoting the distance from the upstream boundary along the streamline. After obtaining all drift coordinates and their gradients (numerically or analytically), one can obtain the gust velocity in regions where the RDT assumptions are valid as follows:

$$\boldsymbol{u}^{(g)} = \nabla X_j u_j^{(\infty)}(X_2, X_3, t - X_1/U_\infty), \tag{22}$$

where $u_i^{(\infty)}$ refers to the undistorted velocity fluctuation at the upstream boundary. Given the upstream fluctuating velocity as a summation of harmonic waves as

$$\boldsymbol{u}^{(\infty)} = \sum_k \Re\left(\hat{\boldsymbol{u}}^{(\infty)} e^{-i\omega t}\right) = \sum_k \Re\left(\hat{\boldsymbol{a}} e^{i\boldsymbol{k}\cdot\boldsymbol{x}} e^{-i\omega t}\right), \tag{23}$$

the distorted gust velocity can also be written in the wavenumber domain as follows:

$$\boldsymbol{u}^{(g)} = \sum_k \Re\left(\hat{\boldsymbol{u}}^{(g)} e^{-i\omega t}\right) = \sum_k \Re\left(\nabla X_j \hat{a}_j e^{i\boldsymbol{k}\cdot\boldsymbol{X}} e^{-i\omega t}\right), \tag{24}$$

where $\hat{a}_i$ is the complex amplitude of the gust velocity at each wavenumber vector $k_i$, and $\omega = k_1 U_\infty$. Note that, at the upstream boundary, $\boldsymbol{x}$ and $\boldsymbol{X}$ are equal to each other.

The next step is to introduce the velocity potential, $\varphi$, which represents the irrotational part of the disturbance field. In practice, $\varphi$ can be expressed in terms of harmonic components, which allows the governing wave equation to be solved for the particular solution associated with the gust velocity[2]. Accordingly, if $\varphi$ is defined as

$$\varphi = \sum_k \Re\left(\hat{\varphi}e^{-i\omega t}\right) = \sum_k \Re\left(\hat{Q}e^{i\boldsymbol{k}\cdot\boldsymbol{X}}e^{-i\omega t}\right), \tag{25}$$

the substantial derivative term ($\mathrm{D}_0^2\varphi/\mathrm{D}t^2$) vanishes such that the wave equation given in Eq. (19) becomes

$$\nabla^2\hat{\varphi} = -\nabla\cdot\hat{\boldsymbol{u}}^{(g)}. \tag{26}$$

Here, the Laplace and the divergence operators can be found as follows:

$$\nabla\hat{\varphi} = i\nabla(\boldsymbol{k}\cdot\boldsymbol{X})\hat{Q}e^{i\boldsymbol{k}\cdot\boldsymbol{X}}$$
$$= ik_j\nabla X_j\hat{Q}e^{i\boldsymbol{k}\cdot\boldsymbol{X}}$$
$$= i\boldsymbol{\kappa}\hat{Q}e^{i\boldsymbol{k}\cdot\boldsymbol{X}},$$
$$\nabla^2\hat{\varphi} = (ik_j\nabla^2 X_j + i\boldsymbol{\kappa}i\boldsymbol{\kappa})\hat{Q}e^{i\boldsymbol{k}\cdot\boldsymbol{X}}$$
$$= (ik_j\nabla^2 X_j - |\boldsymbol{\kappa}|^2)\hat{Q}e^{i\boldsymbol{k}\cdot\boldsymbol{X}},$$

$$\nabla\cdot\hat{\boldsymbol{u}}^{(g)} = (i\boldsymbol{\kappa}\cdot\nabla X_k + \nabla^2 X_k)\hat{a}_k e^{i\boldsymbol{k}\cdot\boldsymbol{X}}, \tag{27}$$

where $\boldsymbol{\kappa} = \nabla X_j k_j$, representing the distorted wavenumber domain. Eventually, the magnitude of the velocity potential term, $\hat{Q}$, is obtained by solving Eq. (26) as

$$\nabla^2\hat{\varphi} = -\nabla\cdot\hat{\boldsymbol{u}}^{(g)}$$
$$(ik_j\nabla^2 X_j - |\boldsymbol{\kappa}|^2)\hat{Q}e^{i\boldsymbol{k}\cdot\boldsymbol{X}} = -(i\boldsymbol{\kappa}\cdot\nabla X_k + \nabla^2 X_k)\hat{a}_k e^{i\boldsymbol{k}\cdot\boldsymbol{X}};$$

$$\hat{Q} = \frac{(i\boldsymbol{\kappa}\cdot\nabla X_k + \nabla^2 X_k)\hat{a}_k}{|\boldsymbol{\kappa}|^2 - ik_j\nabla^2 X_j}. \tag{28}$$

Under the high-frequency gust assumption mentioned above, the expression is simplified by neglecting high-order terms such that

$$\hat{Q} \approx \frac{i\boldsymbol{\kappa}\cdot\nabla X_k\hat{a}_k}{|\boldsymbol{\kappa}|^2}. \tag{29}$$

---

[2]In the present formulation, only this particular solution is retained, while the homogeneous correction that satisfies the exact surface boundary condition is not considered.

Then, the amplitude of the distorted velocity can be found as

$$\hat{\boldsymbol{u}} = \nabla\hat{\varphi} + \hat{\boldsymbol{u}}^{(g)} \tag{30}$$

$$= \left( i\boldsymbol{\kappa}\frac{i\boldsymbol{\kappa}\cdot\nabla X_k}{|\boldsymbol{\kappa}|^2} + \nabla X_k \right)\hat{a}_k e^{i\boldsymbol{k}\cdot\boldsymbol{X}}$$

$$= \left( -\boldsymbol{\kappa}\frac{\boldsymbol{\kappa}\cdot\nabla X_k}{|\boldsymbol{\kappa}|^2} + \nabla X_k \right)\hat{u}_k^{(\infty)},$$

which can also be expressed in terms of a distortion tensor, $D_{ij}$, as

$$\hat{u}_i(\boldsymbol{\kappa}) = D_{ij}\hat{u}_j^{(\infty)}(\boldsymbol{k}), \tag{31}$$

where

$$D_{ij} = \frac{\partial X_j}{\partial x_i} - \frac{\kappa_i \kappa_k}{|\boldsymbol{\kappa}|^2}\frac{\partial X_j}{\partial x_k}. \tag{32}$$

For use in spectral models, as in Amiet's formulation, it is better to compute the distorted energy spectrum directly, which was provided by Glegg and Devenport (2017) through the same distortion tensor as

$$\phi_{ij}(\boldsymbol{\kappa}) = D_{ip}D_{iq}\phi_{pq}^{(\infty)}(\boldsymbol{k}), \tag{33}$$

where $\phi_{ij}$ represents the 3-D energy spectrum. Then, the 1-D spectrum is obtained by integrating over $k_2$ and $k_3$, to be used in the current Amiet's tool.

In this study, to remove the high-frequency assumption and account for all eddy sizes, a more general form of the distortion tensor is derived by directly substituting Eq. (28) in Eq. (30) as follows:

$$\hat{\boldsymbol{u}} = \nabla\hat{\varphi} + \hat{\boldsymbol{u}}^{(g)}$$

$$= \left( i\boldsymbol{\kappa}\frac{(i\boldsymbol{\kappa}\cdot\nabla X_k + \nabla^2 X_k)}{|\boldsymbol{\kappa}|^2 - ik_j\nabla^2 X_j} + \nabla X_k \right)\hat{a}_k e^{i\boldsymbol{k}\cdot\boldsymbol{X}}$$

$$= \left( \frac{-|\boldsymbol{\kappa}|^2\boldsymbol{\kappa}(\boldsymbol{\kappa}\cdot\nabla X_k) - k_j\nabla^2 X_j\boldsymbol{\kappa}\nabla^2 X_k + i\left(|\boldsymbol{\kappa}|^2\boldsymbol{\kappa}\nabla^2 X_k - k_j\nabla^2 X_j\boldsymbol{\kappa}(\boldsymbol{\kappa}\cdot\nabla X_k)\right)}{|\boldsymbol{\kappa}|^4 + (k_j\nabla^2 X_j)^2} + \nabla X_k \right)\hat{a}_k e^{i\boldsymbol{k}\cdot\boldsymbol{X}}. \tag{34}$$

Then, the new distortion tensor becomes

$$D_{ij} = \frac{\partial X_j}{\partial x_i} - \frac{1}{|\boldsymbol{\kappa}|^4 + (k_k\nabla^2 X_k)^2}\left\{ |\boldsymbol{\kappa}|^2\kappa_i\kappa_k\frac{\partial X_j}{\partial x_k} + k_k\nabla^2 X_k\kappa_i\frac{\partial^2 X_j}{\partial x_k^2} \right.$$

$$\left. -i\left( |\boldsymbol{\kappa}|^2\kappa_i\frac{\partial^2 X_j}{\partial x_k^2} - k_k\nabla^2 X_k\kappa_i\kappa_k\frac{\partial X_j}{\partial x_k} \right) \right\}. \tag{35}$$

While more complex in form, this derived tensor remains computationally efficient, as all required derivatives are precomputed. It should be mentioned that Goldstein's RDT, in the present form, does not incorporate the blockage mechanism. This limitation arises because only the scattering potential is solved, and the exact surface boundary condition is not enforced. In this way, a practical, closed-form solution of the RDT is obtained. This omission mainly affects the lowest frequencies, which are outside the range of interest for the present WT applications. Indeed, the present form of Goldstein's RDT includes the vorticity

deflection mechanism of the classical RDT approach. Here, the physical interpretation of the high-frequency approximation
is that the size of the disturbances is assumed to be smaller than the distortion length scale, $l_{\text{dist}}$, which is on the order of
the characteristic length of the body. Mathematically, this implies $kl_{\text{dist}} \gg 1$, where $k$ is the corresponding gust wavenumber.
However, in WT applications, atmospheric eddies often have length scales larger than blade chords, challenging the high-frequency gust restriction. This motivates the use of the generalized version of the tensor (derived without any frequency
restriction) in Sect. 5, where IT noise generation from WTs is predicted by Amiet's model based on RDT. In view of the above,
given an upstream undistorted energy spectrum, the distorted spectrum can be computed using Eqs. (33) and (35), within the
region where the RDT assumptions are valid.

### 4.2 Verification of the generalized distortion tensor

The derived distortion tensor is first verified in a wind tunnel problem where the incoming incompressible flow is axisymmetrically contracting or expanding. In this case, the gradient tensor of the drift coordinates is known as

$$\frac{\partial X_j}{\partial x_i} = \begin{pmatrix} 1/\text{AR} & 0 & 0 \\ 0 & \sqrt{\text{AR}} & 0 \\ 0 & 0 & \sqrt{\text{AR}} \end{pmatrix}, \tag{36}$$

where AR is the area ratio between the inlet and the outlet. Here, $\text{AR} < 1$ refers to contraction (mean flow acceleration),
whereas $\text{AR} > 1$ means expansion (mean flow deceleration).

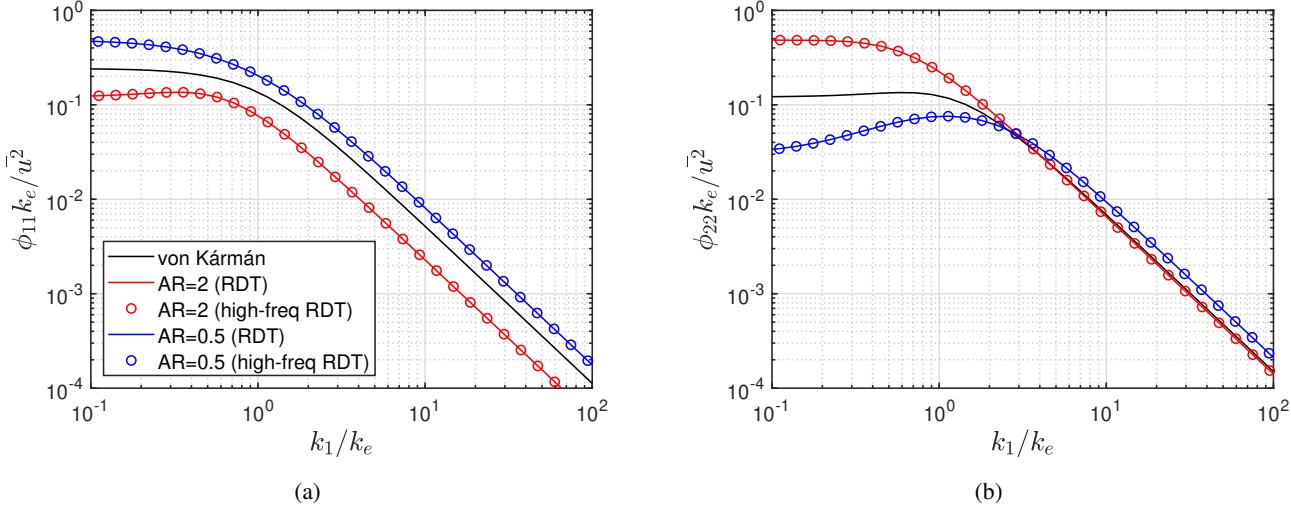

**Figure 8.** 1-D energy spectra of turbulence distorted by axisymmetric wind tunnel contraction ($\text{AR} = 0.5$) and expansion ($\text{AR} = 2$): (a) for
streamwise and (b) for normalwise spectra.

1-D spectra of streamwise and normalwise velocity fluctuations distorted in a wind tunnel are computed by the RDT for both
contraction and expansion cases, and they are shown in Fig. 8. For comparison, the same spectra obtained by using the high-

480 frequency approximated distortion tensor (hereafter, called as high-freq RDT) by Glegg and Devenport (2017) are included as well. The results match exactly, as expected, since all second derivatives of drift coordinates are zero, yielding that both distortion tensors become identical to each other.

The derived distortion tensor is further tested in a problem of a 2-D incompressible turbulent flow approaching a circular cylinder, where experimental data close to the cylinder LE is available (Britter et al., 1979). This benchmark case is particularly
important because the distortion mechanism of a WT blade section is modeled with a representative circular cylinder in this study, which is described in Sect. 5.

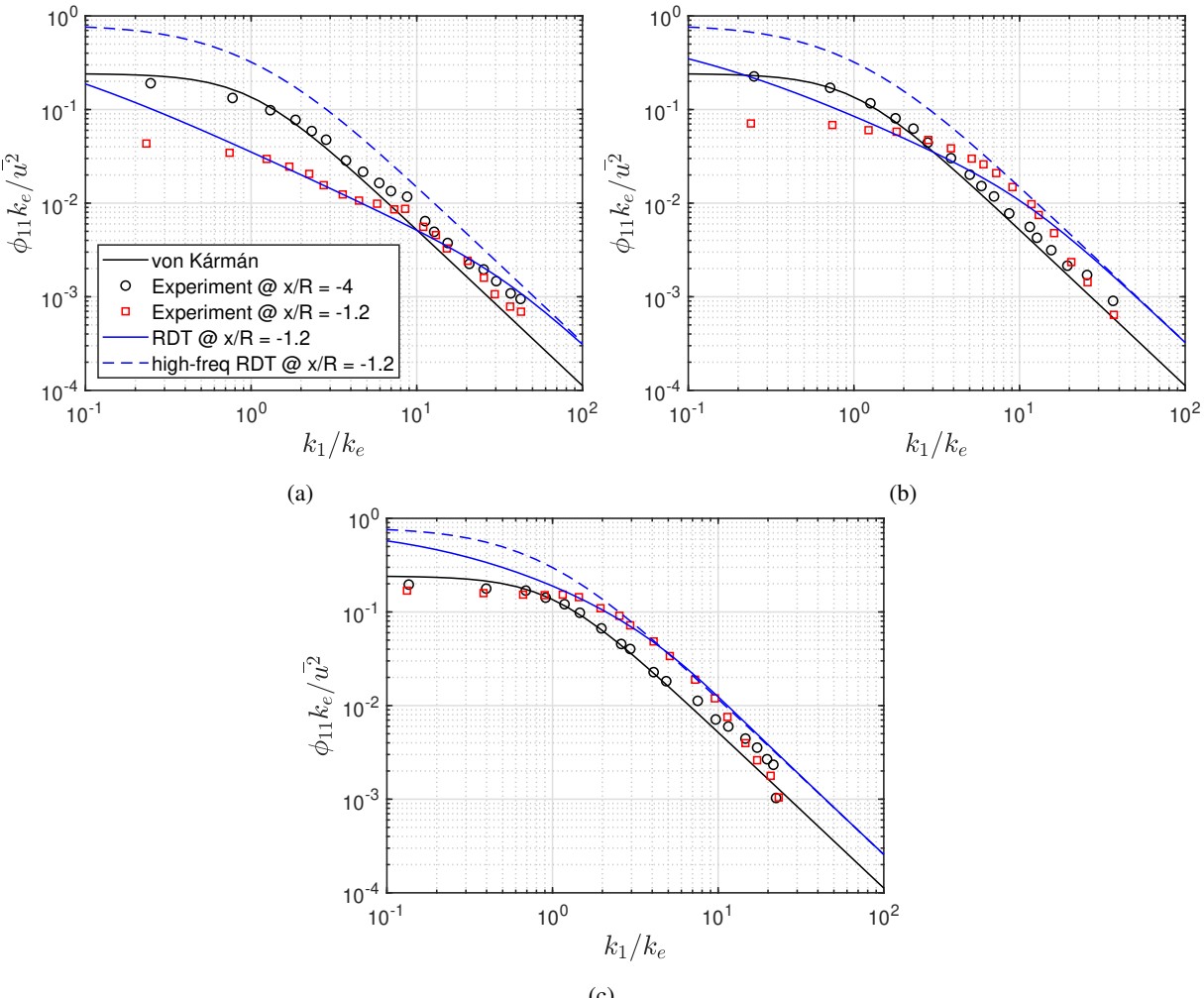

**Figure 9.** 1-D streamwise energy spectra of turbulence distorted by a circular cylinder, shown for different $L/R$: (a) for $L/R = 9.09$, (b) for $L/R = 2.86$, and (c) for $L/R = 1.56$. Experimental data is obtained from Britter et al. (1979).

The incoming flow has a turbulent intensity of $5\%$ approximately. The RDT computations are conducted at an upstream location of $x/R = -1.2$ for three different turbulent length scale ($L$) to cylinder radius ($R$) ratios ($L/R = [9.09, 2.86, 1.56]$). The 3-D vK model is used to estimate the undistorted energy spectra, which allow the computation of the 1-D distorted ones via the RDT. Figure 9 compares measured and predicted streamwise spectra, along with results from the high-freq RDT. The experimental data available at $x/R = -4.0$ and the 1-D vK isotropic spectra are also plotted as references to the undistorted turbulence.

The results show that the vK model accurately matches the undistorted flow at $x/R = -4.0$, providing that the flow is out of distortion effects due to the presence of the cylinder at that position. At $x/R = -1.2$, both RDT formulations capture high-frequency distortion in small eddies ($k_1/k_e \gg 1$) due to vorticity deflection, following the typical RDT mechanisms in the asymptotic limits. However, the RDT with the generalized form of the high-frequency distortion tensor (hereafter, simply denoted by RDT) shows better agreement in the mid-frequency range ($1 < k_1/k_e < 10$), particularly for larger eddies. Notably, the high-freq RDT, in contrast to the RDT, does not vary with $L/R$ and fails to capture the changes through lower frequencies.

The superiority of the generalized version arises because it depends explicitly on the magnitude of wavenumber in the distortion tensor, making the turbulence–geometry interaction sensitive to the turbulence length scale relative to the body size. In this way, the distortion becomes sensitive to the characteristic dimension of the eddies compared to the obstacle radius. In contrast, the high-frequency approximation treats all modes as short waves, and thereby filtering out large eddies (low wavenumbers). This suppresses any dependence on $L$ and leads to inaccurate mid-frequency predictions. In the context of WTs, IT noise generally corresponds to a wider frequency range ($k_1/k_e > 1$), justifying the use of the generalized version in subsequent predictions.

## 5 Analysis of RDT-based IT noise predictions: An evaluation of future WTs

The RDT is integrated into the Amiet-based IT noise prediction tool (called Amiet-RDT hereafter) by replacing the upwash energy spectrum in Eq. (12) with its distorted counterpart. The RDT computations are performed at locations near the stagnation point of each blade profile. The distortion mechanism is applied by approximating the airfoil with a representative circle whose radius is the distortion length ($l_{\text{dist}}$) of the corresponding airfoil.

$l_{\text{dist}}$ can be defined as a derived geometric measure representing the arc distance between the stagnation point and the point of maximum curvature. This definition follows the recent findings of Piccolo et al. (2024, 2026), who showed that thick airfoils distort turbulence in a manner similar to a cylinder with an equivalent radius determined from this geometric measure. This approach avoids the need to recompute the drift coordinates for each airfoil profile, making the RDT implementation computationally efficient. Another perspective has been recently provided by Santos et al. (2024), who proposed an alternative equivalent-cylinder definition based on the average airfoil thickness between the LE and the location of maximum thickness. They demonstrated experimentally that this average thickness reproduces the flow kinematics in the stagnation region more accurately than using either the LE radius or the maximum thickness alone.

Both approaches yield distortion lengths of comparable order and similar scaling with airfoil thickness. Following these insights, and for practical implementation within Amiet-RDT, $l_{\text{dist}}$ is here approximated as a geometric scale lying between the LE radius and half of the maximum thickness, which provides a consistent estimate across blade sections. Since the objective of the present study is to examine relative trends in distortion as blade thickness increases, rather than determine an exact cylinder equivalence for a specific airfoil, the chosen approximation provides a consistent and robust parameterization.

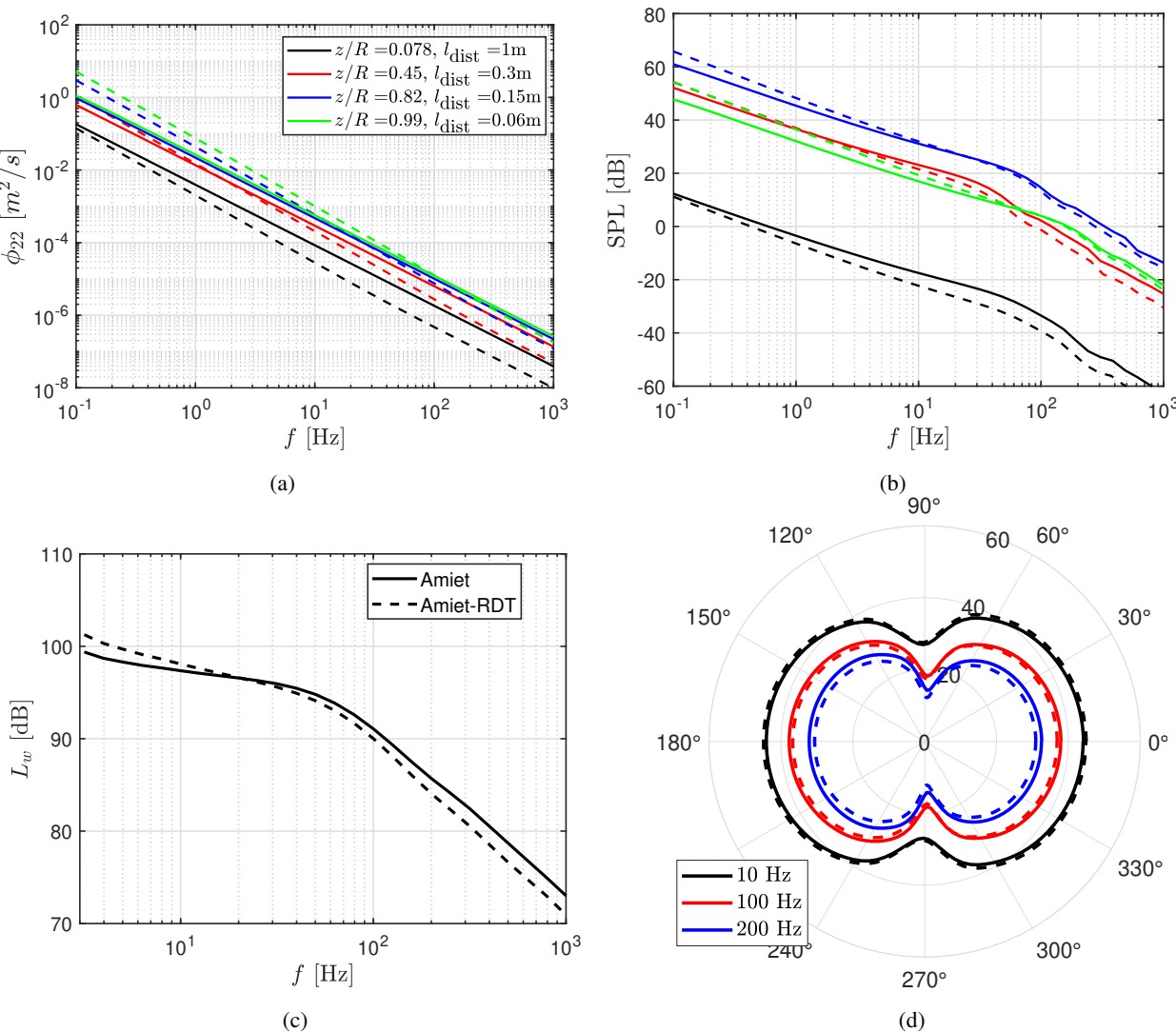

**Figure 10.** Impact of RDT-based distortion on IT noise prediction for the benchmark WT configuration used in Sect. 2.2. (a) and (b) show upwash velocity and SPL spectra obtained at different blade sections. (c) compares WT sound power level spectra. (d) demonstrates SPL directivity at different frequencies. Solid lines refer to the calculations based on the vK model whereas dashed lines refer to the ones with RDT.

The developed Amiet-RDT tool is applied to the benchmark WT configuration used in Sect. 2.2 to assess how turbulence distortion affects the IT noise spectrum. The distorted spectra are computed at points located at $x/l_{\text{dist}} = -1.05$ upstream of each segment leading edge. Case 1 is used for this analysis, and the results are compared with those obtained by the Amiet tool, which is based on the vK isotropic model. The upwash spectra at $x/l_{\text{dist}} = -1.05$ are shown in Fig. 10 (a) for different blade sections, the span location of which is specified by $z/R$, $R$ being the rotor radius. The corresponding distortion lengths are also given. From root to tip, $l_{\text{dist}}$ decreases, as expected from typical blade profile thicknesses. The results indicate that, within the low-frequency range of interest for IT noise, the distorted spectra are not significantly different from the isotropic ones. This is because the relevant frequencies fall within the transition region between the two RDT asymptotic limits. As a result, the sound pressure level (SPL) contribution from each section remains almost unchanged, as shown in Fig. 10 (b). An exception is observed close to the blade root, which has a circular profile and the largest $l_{\text{dist}}$. A larger $l_{\text{dist}}$ increases the range of high-frequency eddies affected by distortion, leading to more small-scale structures with reduced energy. However, these inner blade regions contribute minimally to overall noise, so the total generated IT noise (quantified in terms of sound power level, $L_w$, in Fig. 10 (c)) is predicted nearly identically from both Amiet and Amiet-RDT models. The typical IT noise directivity pattern on the horizontal ground plane, shown in Fig. 10 (d), where 0 degree corresponds to the observer location, confirms this result as well.

To assess the potential impact of turbulence distortion in next-generation, larger WTs, the same analysis is repeated for a conceptual WT scaled to reflect the anticipated growth in onshore turbine sizes (Veers et al., 2019). In this configuration, the rotor radius of the benchmark turbine is tripled, and the blade chords are scaled linearly with the span. Although industry practice suggests that chord growth tends to be more than linear (i.e., growing faster than proportional to the change in span size) near the root and less than linear near the tip (Bortolotti et al., 2016, 2019), this parametric study focuses on relative changes in turbulence distortion with increasing thickness. Hence, a uniform geometric scaling of the chord provides a plausible first-order assumption (Canet et al., 2021).

For the blade thicknesses, however, a more than linear scaling is applied. For very large turbines, designers tend to increase relative thickness to maintain strength and stiffness without excessively increasing the blade mass (Timmer and Bak, 2013; Caboni et al., 2017). In the benchmark case, airfoils at mid-to-outer span have a LE radius of about $0.02c$ and a maximum thickness of about $0.18c$, where $c$ refers to the chord. In the next-generation turbines, LE radii of approximately $0.05c$ and maximum thicknesses of $0.35$–$0.40c$ are expected, according to industry indications and literature (Timmer and Bak, 2013; Schaffarczyk et al., 2024). Accordingly, the ratio of distortion length to chord of each section, except near the root where the circular profile exists, is doubled in the conceptual configuration. Finally, the same flow and operating conditions as Case 1 are retained, except that the RPM is reduced by one-third to preserve comparable relative velocities with respect to the benchmark case.

Figure 11 demonstrates the influence of turbulence distortion on noise predictions for this enlarged turbine, as previously. Corresponding increases in $l_{\text{dist}}$ for the same sections are shown in Fig. 11 (a). The main finding is that thicker blade profiles, under the same inflow conditions, cause a significant distortion of the turbulent velocity field, which is observed by the alteration of the corresponding spectrum within the frequency range of interest (see Fig. 11 (a)). Across all blade sections, the

distortion lies in the high-frequency regime of RDT, where vorticity line deflections reduce the energy of smaller eddies. As a result, thicker profiles yield reduced IT noise, a trend clearly visible in both the overall sound power levels and directivity patterns, as shown in Figs. 11(c-d). This reduction increases with frequency. When compared with the benchmark configuration in Fig. 11 (c), the standard Amiet model predicts a monotonic rise in noise with rotor size, whereas the Amiet-RDT tool shows that spectral alterations mitigate this increase. At higher frequencies, the larger turbine can even produce noise levels comparable to the benchmark configuration.

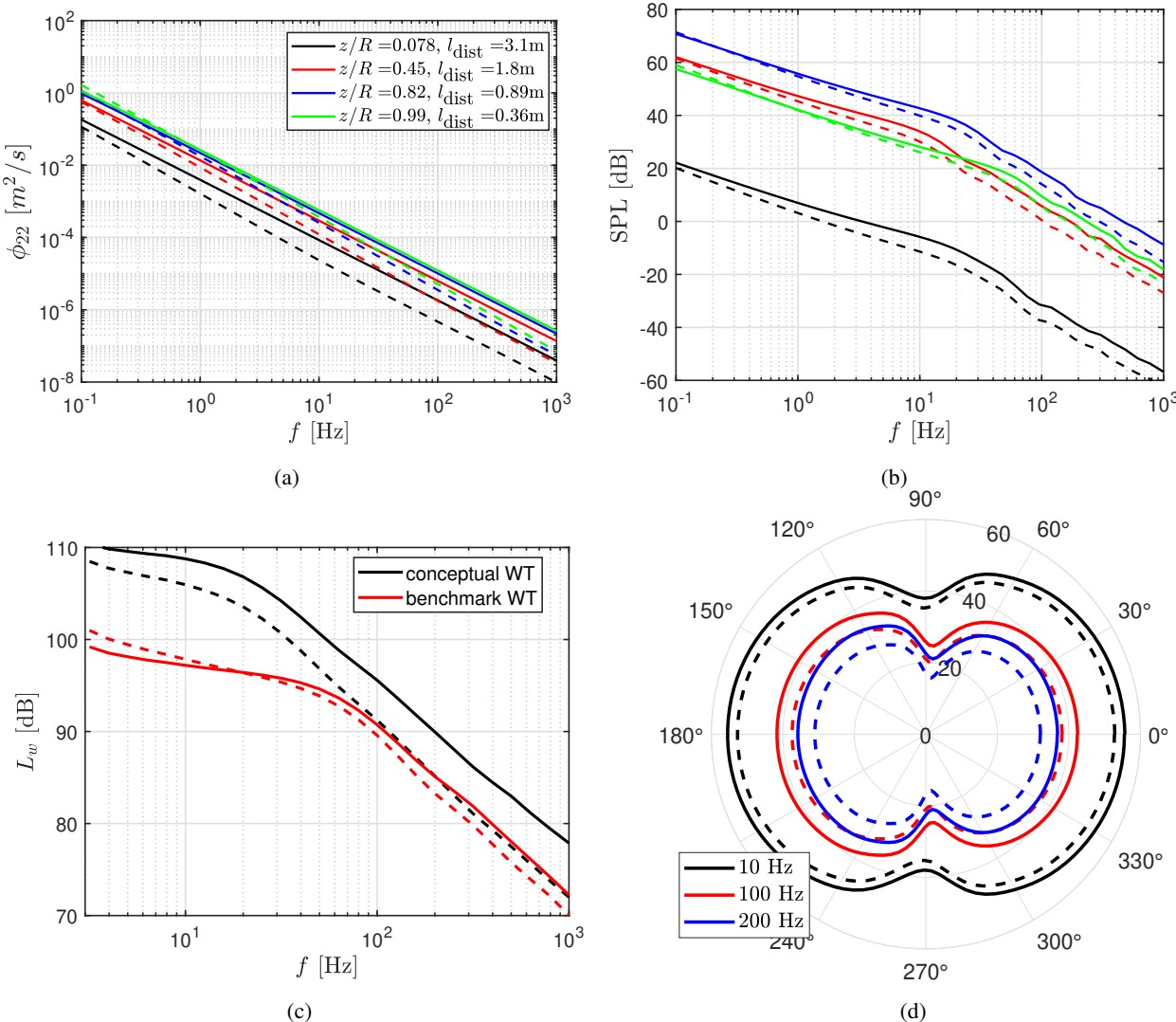

**Figure 11.** Impact of RDT-based distortion on IT noise prediction for the conceptual WT configuration. (a) and (b) show upwash velocity and SPL spectra obtained at different blade sections. (c) compares sound power level spectra for different WTs. (d) demonstrates SPL directivity at different frequencies. Solid lines refer to the calculations based on the vK model whereas dashed lines refer to the ones with RDT.

It should be noted that the present analysis specifically examines IT noise in order to clarify the role of turbulence distortion near the blade surface. In real WT operations, however, IT noise coexists with other broadband mechanisms, especially TE noise. When TE noise is included in the predictions, its contribution to the noise generation may partially mask turbulence distortion effects on noise, particularly at frequencies above approximately $100\,\mathrm{Hz}$. As a result, the impact of turbulence distortion identified in this study may not be directly apparent in overall noise metrics unless noise components are examined separately or frequency-resolved analyses are performed. Nevertheless, the present results remain relevant for understanding how turbulence distortion redistributes spectral energy, especially at low frequencies where IT noise is the dominant contributor for large turbines with thick blades.

The above analysis isolates thickness effects while keeping the inflow and operating conditions the same. Figures 12(a–b) demonstrate the same analysis by varying the incoming turbulence length scale for the conceptual rotor. For this case, the wind speed and the turbine rotational speed are set to $6\,\mathrm{m\,s^{-1}}$ and 13 RPM, respectively. At higher frequencies, corresponding to smaller eddies, noise levels scale inversely with the length scale since this quantity mainly reflects the energetic large eddies, as also reported by Botero-Bolívar et al. (2024). Within this regime, the Amiet-RDT model consistently predicts lower noise than the baseline Amiet approach. Changes in length scale also shift the spectral shape, which is further influenced directly by turbulence distortion. As a result, the Amiet-RDT model can even predict higher low-frequency noise, as observed for $L = 10\,\mathrm{m}$.

Another scenario is studied by fixing the length scale to $L = 10\,\mathrm{m}$ and varying the rotational speed between 1 and 16 RPM. The results are shown in Figs. 12(c-d). Within the IT noise frequency range, the predicted noise levels increase proportionally to the rotational speed, in contrast to the previous case. Again, due to the change in spectral shape induced by the distortion effect, the differences between the Amiet and Amiet–RDT predictions depend on both frequency and speed. In particular, the distortion effect caused by the thicker WT profiles is found to be more evident at lower rotational speeds.

These results imply that IT noise levels do not necessarily scale proportionally with rotor size once realistic blade thickening is accounted for in future WTs. Turbulence distortion changes the balance across frequencies, reducing high-frequency noise but sometimes amplifying low-frequency contributions. Consequently, trends predicted with the canonical spectra or the empirical models based on certain frequency ranges may not remain valid for future large-scale designs.

## 6  Conclusions

This study investigates the relevance of IT noise for next-generation large WTs by explicitly accounting for turbulence distortion effects considering their growing rotor sizes, an aspect overlooked in existing low-fidelity approaches relying on empirical corrections. In this sense, an accurate, computationally efficient, and physics-based IT noise prediction tool based on Amiet's analytical model is developed. Two main mechanisms are examined: streamwise spectral changes in the induction zone and distortion near thick blade LEs.

The turbulence distortion due to streamtube expansion in the induction zone is examined by LES coupled with modeling the blade forces via SOWFA. The spectral results reveal that turbulence distortion in the induction region remains negligible

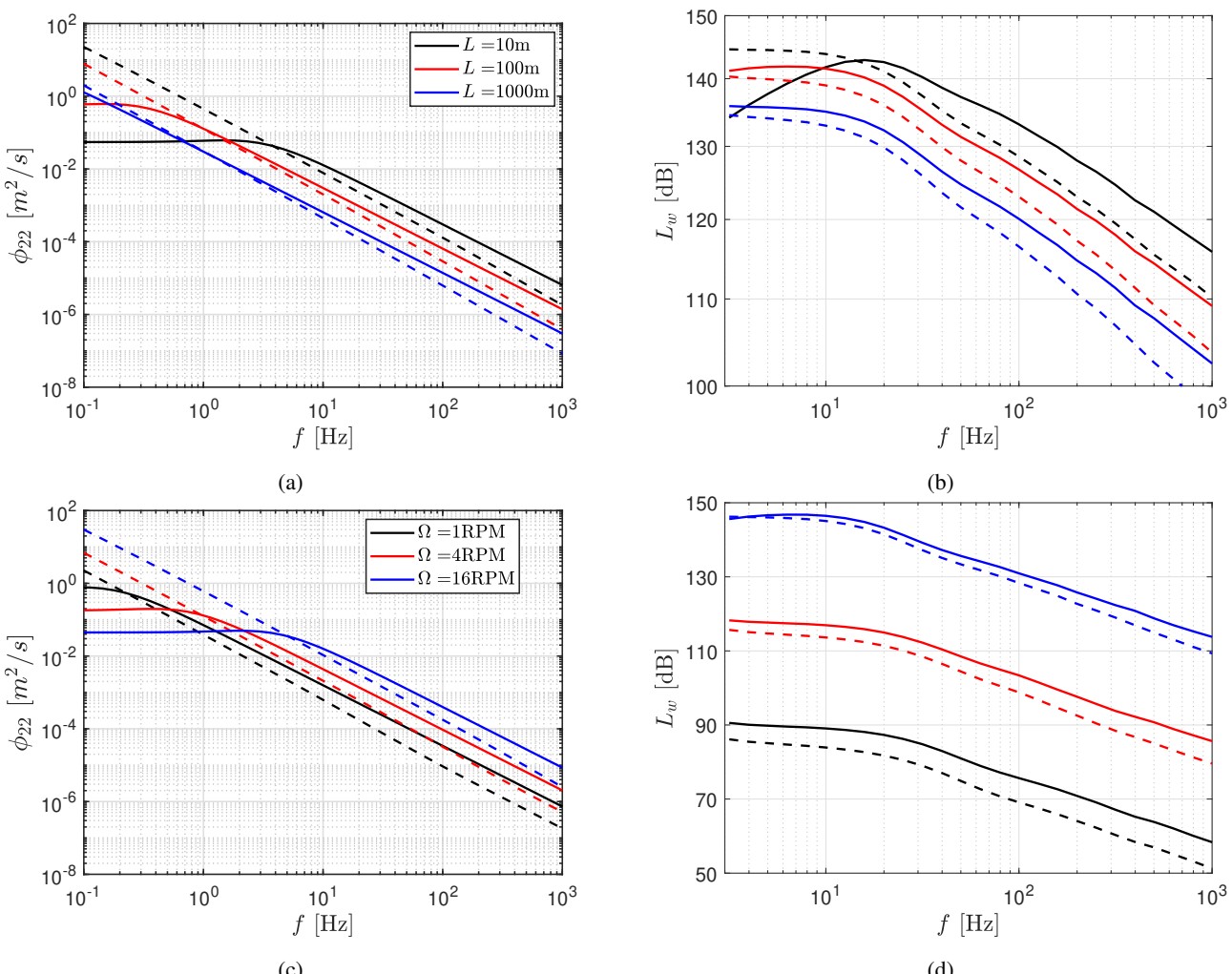

**Figure 12.** Assessment of IT noise predictions with and without RDT for different turbulent length scales (a-b) and different turbine rotational speeds (c-d). (a) and (c) show upwash velocity spectra at $z/R = 0.8$ while (b) and (d) show WT sound power level spectra. Solid lines refer to the calculations based on the vK model whereas dashed lines refer to the ones with RDT.

in the mid-to-high frequency range, which is relevant for acoustic scattering. This result suggests that the upstream turbulence spectra estimated by isotropic models can be used for this region without reducing accuracy.

To address distortion due to blade thickness that appears as the upstream turbulence approaches the blade surfaces, a practical form of Goldstein's RDT, which is associated with vorticity deflections, is developed. Unlike earlier approaches, this model does not rely on the high-frequency gust approximation and remains valid across a broad frequency range within its capabilities. It is implemented into an Amiet-based IT noise tool through a distortion tensor formulation, which directly modifies the upwash

spectrum near the blade LEs. The method is validated against canonical test cases, and its accuracy in capturing the vorticity deflection effects of the RDT mechanism is demonstrated.

Importantly, the developed Amiet-RDT tool modifies the input turbulence spectrum based on geometry (via a representative distortion length), operational conditions, and inflow properties. This eliminates the need for empirical coefficients commonly used to adjust isotropic energy spectrum models, which are not suitable for realistic scales of atmospheric turbulence and WTs. This tool increases prediction accuracy without additional computational cost, making the developed model practical for parametric noise assessments. It is believed that this model will be especially beneficial in the design and development of WT noise mitigation strategies.

When applied to a conceptual onshore WT scaled through an existing benchmark configuration, the developed tool reveals that turbulence distortion near blade surfaces becomes increasingly influential for future turbines. This is particularly the case as relative thicknesses are expected to scale more than linearly - reaching around $0.4c$ from the current $\sim 0.2c$ - to maintain structural strength without excessive mass growth. The results show that IT noise levels exhibit nonlinear behavior with rotor size, contrary to what is predicted by standard models. The distortion effects, particularly the suppression of small-scale eddy energy by vorticity deflection, can even reduce overall noise generation at certain frequencies for a larger rotor where higher noise would normally be expected based on geometric scaling alone. The results also indicate that turbulence distortion interacts with the inflow length scale and rotational speed, leading to frequency-dependent shifts in IT noise levels. This reinforces that scaling trends in the noise spectrum cannot be generalized without accounting for both geometric and operational factors. It is also emphasized that, when other mechanisms such as TE noise are included in the predictions, the relative impact of these turbulence distortion effects on WT noise may be less apparent, particularly at higher frequencies.

As a recommended next step, a high-fidelity simulation of the entire WT flow field, including near-blade domain, can be performed. This approach would allow a detailed investigation of near-blade turbulence distortion and provide valuable insight regarding possible rotor-induced distortion effects at wider frequency ranges. Such work could serve to further validate the RDT-based predictions.

*Code and data availability.* The Amiet–RDT MATLAB tool (Yalçın and Piccolo, 2026) for wind turbine inflow turbulence noise predictions presented in this study is openly available on Zenodo at https://doi.org/10.5281/zenodo.18198345. The repository includes the source code and example input files required to reproduce some of the results shown in this paper. All the remaining data that support the findings of this study are available from the corresponding author, ÖY, upon reasonable request.

*Author contributions.* All authors contributed to the conceptualization of the study. ÖY and AP developed the Amiet-based prediction tool. ÖY carried out the SOWFA simulations and subsequent postprocessing. ÖY, AP, and RZ contributed to the investigations and derivations of the RDT formulation. ÖY performed the data analysis, visualization, and validation of the results. DR and RM acquired the funding and coordinated the project. ÖY wrote the manuscript, and all authors contributed to its reviewing and editing.

*Competing interests.*    The authors declare that they have no conflict of interest.

*Acknowledgements.*    This paper is part of the Blade Extensions for Silent Turbines (BEST) project (HER+22-02-03415120) which is supported by the Netherlands Enterprise Agency (Rijksdienst voor Ondernemend Nederland (RVO)), acting on behalf of the Dutch Ministry of Economic Affairs, using the Renewable Energy Transition subsidy (subsidie Hernieuwbare Energietransitie (HER+ 2022-02)), part of the Topsector Energy (TSE) subsidies.

The authors gratefully acknowledge Dr. Dachuan Feng and Nirav Dangi for their support with running the SOWFA simulations on the DelftBlue cluster.

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
