# Peer review of "Turbulence Distortion Matters in Predicting Inflow Turbulence Noise of Future Wind Turbines"

_Wind Energy Science, 2025_

## Author Response (AR1)

**Response to Reviewer Comments**

**To Reviewer 1**

We thank the reviewer for the careful and constructive feedback. We have carefully addressed all comments, as listed below respectively with our responses (RC: Reviewer's comment, AC: Authors' comment). Changes have been incorporated throughout the revised manuscript and highlighted in red.

1) RC: One is related to the terminology used to describe the change in slope for the turbulence spectra. It is formulated in p.5 and p.16 that the slope (e.g. of -5/3) is "reduced", when the reviewer would rather in both case that the slope is increased. The reviewer assumes here that the authors mean that the coefficient of the logarithmic slope is increased. It is possibly a matter of taste about how to formulate this information.

AC: We agree with the reviewer that the terminology is ambiguous. The slope becomes more negative, but its magnitude increases, so it is indeed getting steeper. We have revised the corresponding words on p.5 and p.16.

2) RC: The length scale of 312 m appears quite large to the reviewer. At which altitude is it evaluated?

AC: The integral length scale is evaluated at hub height (80 m), which is given in Section 2.2. We have clarified this explicitly in the text.

3) RC: There exist an inconsistency in Fig.3. Fig.3(f) is a zoom-in of Fig.3(c), however in 3(c) the model results overestimate the measurements in the frequency range 40-100 Hz, while it is the opposite in 3(f).

AC: We appreciate the reviewer catching this. Figure 3(d) is mistakenly copied into Fig. 3(f). We have renewed Fig. 3(f).

4) RC: p.14, l.325: It is not clear to the reviewer why it can be concluded that the turbulence is isotropic.

AC: The sampled turbulence energy spectrum collapses onto a vK isotropic spectrum, up to the cut-off frequency. This vK spectrum is estimated using the sampled turbulence characteristics. We have clarified the reasoning in the text.

5) RC: p.16, l.360: A Blade-Passage-Frequency of 0.45Hz is consistent with 9 RPM. However, in Fig. 7, the first spectral peak is located at 0.9Hz... which is inconsistent.

AC: We thank the reviewer for noticing this. When analyzing the data from probes closer to the turbine, the harmonic peaks, including the first one, appear more clearly. So, this is a matter of upstream distance. To make the discussion in the text clearer, we have renewed Fig. 7 by including the data at -0.125D.

6) RC: Furthermore, in Fig.7 the slope "decrease" above, say, 0.1 Hz, which corresponds to the mesh cut-off frequency in Fig.4, is not clearly explained. In the reviewer's opinion, the low resolution upstream of the refined mesh zone contains large scale turbulence. During the time period for the turbulent flow to reach the measurement point, e.g. at -0.25D, smaller structures don't have the time to develop through the energy cascade of turbulence, which could explain the lower energy level (above 0.1Hz) than expected.

AC: We thank the reviewer for this observation. We agree that the upstream mesh resolution influences the development of small turbulent scales between the cut-off frequencies of the upstream coarse mesh and the refined near-rotor mesh. Importantly, the same behavior appears consistently at all upstream locations, and not only near the rotor, confirming that it is not associated with rotor-induced distortion, which is the central question of this section. In the revised manuscript, we have provided a clearer explanation of this effect.

7) RC: The reviewer is not familiar with the RDT equations and their derivation. However, the argument that the second order Lagrangian derivative disappear from Eq. 19 to give Eq. 29 is not clear for the reviewer.

AC: The velocity potential term ($\varphi$) is defined in such a form that its second-order Lagrangian derivative ($D^2 \varphi / Dt^2$) vanishes, as also explained by Glegg \& Devenport (Aeroacoustics of Low-Mach-Number Flows, 2017, p. 257). This allows Eq. 19 to reduce directly to Eq. 26. The revised text now explains this step more clearly.

8) RC: The phrasing in p.23, l. 498-500 is not clear. The distortion length is first defined as the/a (?) circle radius. Then, it is defined by the "length between the LE radius and half of the max. thickness" which doesn't make sense as it does not describe a length per se... or does the reviewer misunderstand?

AC: Our intention was to give a practical estimation, but we agree that the definition given in the text is ambiguous. We have now clarified this in the manuscript.

Indeed, when we apply RDT, we approximate the airfoil by a representative circle whose radius is chosen to reproduce the distortion induced by this airfoil. This radius is a derived

geometric measure capturing the distance between the stagnation point and the point of maximum curvature. This is consistent with the approach of Piccolo et al. (JSV, 2026), who showed that thick airfoils distort turbulence similarly to a cylinder with an equivalent radius. The revised text now explicitly states that definition, together with a consistent estimate across blade sections for practical implementation.

9) RC: p.23, l.507: The spectra are plotted for a position in the vicinity of the LE, but it is not specifically defined.

AC: We agree that the positions where we apply RDT should be explicitly given. These points are located at $x/l_{dist}=-1.05$ upstream of each airfoil leading edge. We have added this information in the revised manuscript.

**To Reviewer 2**

We thank the reviewer for the careful and constructive feedback. We have carefully addressed all comments, as listed below respectively with our responses (RC: Reviewer's comment, AC: Authors' comment). Changes have been incorporated throughout the revised manuscript and highlighted in red.

Specific comments:

1) RC: In line 290, the authors mention the turbulent length scale and turbulence intensity obtained from simulations. As these parameters are crucial for the TI noise predictions, more details should be provided regarding how they were obtained, such as the type of simulation performed, how the turbulent length scale was calculated / determined, the location (altitude) at which the parameters were obtained, and whether this altitude is representative of the wind turbine test case (is it at the centre of the wind turbine rotor?).

AC: We agree that the inflow parameters should be described more clearly. In the revised manuscript, we now explicitly state that the turbulence intensity and integral length scale are taken directly from the zEPHYR project. These parameters were obtained from the Weather Research and Forecasting (WRF) coupled with LES simulations performed by Kale (2024), which is a part of the same project. In this study, these parameters were computed at the hub height (80 m) of the same (SWT-2.3-93) reference wind turbine, and the integral length scale was computed using the autocorrelation function based on second-order turbulence statistics obtained from velocity fluctuations. Since our study uses this benchmark case for validation, we considered these recommended values for the reference turbine.

2) RC: In line 291, the authors state that these turbulence quantities were obtained for Case 2 and assumed to remain unchanged for the other cases. How realistic is this? The turbulence intensity and length scale typically vary with the inflow velocity, which differs by about 25\% for the other cases compared with Case 2. Could the authors please elaborate on the justification for this assumption and discuss its potential implications?

AC: The turbulence intensity and integral length scale used in this study were provided only for Case 2 in the zEPHYR benchmark and the associated WRF–LES simulations. This is the standard convention followed in previous Amiet-based wind-turbine noise studies using the same benchmark (e.g., Botero-Bolívar et al., Renewable Energy, 2024), where the same inflow parameters are applied to all operating conditions.

Importantly, our validation against Botero-Bolívar et al.~(2024) relies on using exactly the same turbulence characteristics they used. This ensures a consistent baseline for comparing IT noise predictions when the inflow energy spectra are evaluated by an isotropic von Kármán model.

We agree with the reviewer that the turbulence intensity and length scale may vary with the wind speed in reality, causing deviations between predicted and measured noise levels. However, the ZEPHYR benchmark provides only one atmospheric dataset, and its use across operating points is part of the benchmark protocol. We have clarified these points in the revised manuscript.

3) RC: In Section 3.1, it is not clear whether an artificial turbulence generator was used to initially create the turbulence in the flow. Could you please include more details on this matter? Additionally, what were the values of the turbulence intensity and the turbulence length scale? How does the size of the turbulence length scale compare with the rotor? This information is relevant to the reader and should be added to the paper.

AC: In SOWFA, the turbulent flow is not generated using a synthetic turbulence model; instead, it is produced through a prescribed gradient, which is set to be a neutral atmospheric boundary layer before running the precursor simulation. In this precursor run, turbulence develops naturally from shear and surface roughness. We have clarified this point in Section 3.1.

The turbulence intensity and length scale in the induction zone are approximately 10\% and 150 m, respectively, while the rotor diameter of the NREL 5MW turbine is 126 m. In the revised manuscript, these numbers have been explicitly given. The turbulent structures are of comparable or larger size than the rotor, a point that we mentioned during the discussion in Section 3.2 of the manuscript.

4) RC: In Section 3.1, please include the cut-off frequency used for the simulations, as well as any dependence of this frequency on the locations where the energy spectrum was analyzed in Section 3.2. The cut-off frequency should also be reported in Section 3.2 and Section 5.

AC: We thank the reviewer for this suggestion. Following both reviewers' comments, we have added the numerical cut-off frequencies directly in Sections 3.1 and 3.2, noting that the coarse upstream mesh corresponds to a cut-off of approximately 0.1 Hz, while the refined mesh (covering all the analyzed probe points as described in Section 3.2) increases this value to about 1.6 Hz.

Regarding Section 5, this part of the paper is based on analytical formulations to obtain energy spectra and does not involve any simulated spectra; therefore, a numerical cut-off frequency is not applicable. We hope that this clarification addresses the reviewer's concern.

5) RC: Figure 4: How was the turbulence length scale determined, which was used as input to the von Kármán spectrum?

AC: The turbulence length scale was obtained directly from the sampled streamwise velocity during simulations. Specifically, we computed the integral length scale from the power spectral density by evaluating its standard spectral integral definition (i.e., based on the ratio of the integrated energy and its wavenumber-weighted form). This is equivalent to computing the integral scale from the longitudinal autocorrelation function. We have provided a brief explanation of this in Section 3.1.

6) RC: In line 357, the authors mention that the focus will be on mid- to high-frequency ranges. However, the definitions of low, mid, and high frequencies are not clear. Please specify earlier in the manuscript the frequency ranges that the authors consider to represent low, mid, and high frequencies.

AC: We thank the reviewer for pointing out this ambiguity. In the manuscript, the terms "mid-to-high frequencies" of atmospheric turbulence refer to the acoustically relevant frequency range, i.e. the band in which IT noise is generated. This corresponds approximately to 1–200 Hz, commonly classified in the literature as low-frequency noise in wind-turbine acoustics.

By contrast, when discussing atmospheric turbulence, "low frequencies" refer to very large eddy scales whose characteristic frequencies are well below 1 Hz. These scales produce infrasound and do not contribute significantly to audible IT noise.

To avoid confusion, we have clarified these points in Introduction between Lines 167-170 (referring to the initial manuscript) by mentioning:

 - "Low frequencies" of atmospheric turbulence: way below ~1 Hz (large eddies, infrasound).

- "Mid-to-high frequencies" in our analysis: the part of the turbulence spectrum contributing to IT noise, i.e. the 1–200 Hz band.

7) RC: In Section 3.2, the authors conclude that the turbulence distortion due to streamtube expansion is negligible for frequencies relevant to noise generation. Out of curiosity, have the authors analyzed lower frequencies as well? If so, did they observe any distortion of the turbulence at these larger length scales? Additionally, was it possible to simulate very low frequencies (i.e., very large length scales) accurately within the domain used?

AC: Our analysis focused on the frequency range relevant for noise generation (mid-to-high frequencies), and the simulation period was therefore not long enough to accurately resolve the very low frequencies associated with the largest atmospheric eddies. Capturing such frequencies was possible but would require substantially longer simulations (both precursor and wind-turbine-included ones), which was beyond the scope of the present acoustic-focused study.

However, at the beginning of Section 3.2, we analyzed the mean and low-order statistical quantities obtained from simulated velocity data. These reflect the behavior of large eddies whose characteristic sizes exceed the rotor diameter. As discussed in the manuscript, these large scales show only gradual variation and no clear evidence of rotor-induced distortion in the frequency range relevant to noise.

8) RC: In line 500, the authors justify the modelling of an airfoil by a representative circle of a certain diameter, which is a common approach in aeroacoustics. This matter has also been investigated experimentally before (see dos Santos et al., 2024, https://arc.aiaa.org/doi/10.2514/1.J063122), where the authors suggested a different approach for determining the representative cylinder diameter than the one used in the current manuscript. What is the difference between the method applied in this study and the approach proposed in that work? How comparable are the cylinder diameters obtained using both methods? What are the implications of using a different cylinder diameter for the results and conclusions of the present paper? As this parameter ($l\_dist$) is highly relevant to the results presented in the manuscript under review, please discuss the assumptions made and their consequences, in light of the findings from both studies in the literature (dos Santos et al., 2024; Piccolo et al., 2024), including answers to the questions above.

AC: The reviewer raises a valid concern, as this point should indeed be addressed more clearly. In the revised manuscript, a more detailed explanation has been added, clarifying both the objective of the present analysis and the adopted approach.

The purpose of the corresponding section is not to quantify with precision the distortion associated with possible future wind turbine blade sections, but rather to evaluate the

expected noise emission considering current trends in rotor-diameter growth and blade thickening, within a low-fidelity Amiet-RDT framework. For this purpose, a geometrically derived, single-parameter representation of airfoil bluntness is sufficient, as long as it yields a distortion length ($l\_dist$) that scales consistently with airfoil thickness and leading edge radius.

In this study, $l\_dist$ is approximated as a geometric scale lying between the LE radius and half of the maximum thickness, which provides a consistent estimate across blade sections when we scale the zEPHYR turbine to a conceptual enlarged design. This approximation is guided by the findings of Piccolo et al.~(2024,2026), who demonstrated that thick airfoils distort turbulence similarly to a cylinder whose radius is defined by a geometric–curvature-based measure (arc distance between the stagnation point and the point of maximum curvature).

We acknowledge the valuable contribution of dos Santos et al.~(2024), who independently proposed an equivalent-cylinder definition based on the average thickness between the LE and the location of maximum thickness, derived from detailed experimental observations of stagnation region kinematics. Upon revisiting their results (prompted by the reviewer's comment), we recognize that their definition is highly consistent with, and in fact strongly supports, our chosen approximation, as both lead to geometric scales of similar magnitude and physical interpretation.

Regarding comparability, we note that the two definitions (despite differing in geometric rationale) produce distortion lengths of similar order of magnitude and exhibit similar scaling with increasing airfoil thickness. This is now explicitly stated in the revised manuscript. Because the conclusions of this study rely primarily on relative changes in distortion with rotor and thickness scaling - not on the absolute value of $l\_dist$ - the specific definition used has no qualitative influence on our findings. These points have been clearly explained in the revised manuscript at the beginning of Section 5.

9) RC: In Section 5, the authors refer to trends that are "more than linear." Please clarify what is meant by this.

AC: We thank the reviewer for pointing out this ambiguity. We mean that "more than linear" refers to geometric parameters (such as chord or relative thickness) increasing faster than with proportional scaling with the rotor size or span. In the revised manuscript, the phrase "more than linear" has been clarified at its first appearance in Section 5.

10) RC: In Section 5, the reviewer notes the absence of a discussion on the relevance of turbulence distortion for the future wind turbines, taking into account the frequency range at which trailing-edge noise is dominant. The authors mention that the effects of the

turbulence distortion are even more relevant for high frequencies, which are usually dominated by trailing-edge noise. Therefore, a clear definition of what constitutes low and high frequencies should be provided. Additionally, a discussion considering trailing-edge noise should be added, indicating the frequency range in which turbulence distortion is expected to have an impact. This should also be included in the conclusions, specifying the relevant frequency range (with numerical values) and the expected level differences in dB when turbulence distortion is taken into account.

AC: We thank the reviewer for raising this point. We have clarified in the Introduction the definitions of low- and high-frequency ranges in relation to IT noise as well as the atmospheric turbulence, as suggested by the reviewer in Comment 6.

Regarding trailing edge (TE) noise: while we agree that inflow turbulence can influence boundary layer development and therefore may have a secondary effect on TE noise, the dominant mechanism for TE noise remains the boundary layer turbulence and its development along the blade surface. The RDT formulation developed in this work modifies only the upstream inflow turbulence approaching the leading edge; it does not model the evolution of the boundary layer or the near-wall turbulence responsible for TE noise generation. For this reason, extending the discussion to frequency ranges dominated by TE noise would require a different modeling framework and is outside the scope of the present paper. Including a dB-level analysis in the TE noise regime would not be meaningful without a dedicated TE noise model.

Technical corrections:

1) RC: Please recheck the nomenclature table, as some variables appear to be duplicated (e.g., U – mean velocity), and some are unclear. For example, do the variables v (unsteady velocity) and u (fluctuating flow speed) represent the same quantity?

AC: We agree that the nomenclature requires clarification. In the original version, some quantities, especially the velocity terms, were listed multiple times because different sections of the paper used them in different contexts. We have revised the nomenclature to remove duplicates and ensure internal consistency throughout the manuscript. Besides, we have grouped quantities for easier reference.

Regarding the velocity terms in RDT: v is the total unsteady velocity including both the mean (U) and the fluctuating (u) part. So, it is different than u. We here followed the RDT literature.

2) RC: In line 196, the reviewer suggests adding the steps immediately after the colon. Currently, the colon implies that a list will follow, but the authors are referring to the subsubsections instead.

AC: We agree with the suggestion. In the revised manuscript, we have given the steps immediately and continued with the subsubsections.

3) RC: Please include in the caption of Figure 10 a description of what the continuous line represents.

AC: The captions of Figures 10, 11, and 12 have been updated accordingly.

---

## Author Response (AR2)

**Response to the Editor and Reviewers**

We would like to thank the Editor and both reviewers for their careful evaluation of our manuscript and for the constructive comments provided throughout the review process. We are pleased that the paper is considered nearly ready for publication and have addressed the remaining minor comment as requested.

In addition, a new "Code and data availability" section has been added to the manuscript, providing access to the Amiet–RDT MATLAB tool used in this study, which has been recently made openly available via a public Zenodo repository.

**To Reviewer 1**

We thank the reviewer for the positive assessment and appreciate the recommendation for publication of the paper.

**To Reviewer 2**

We thank the reviewer for the positive assessment and the constructive feedback. In response to the remaining comment regarding the potential masking of turbulence distortion effects when trailing edge (TE) noise is included, we have added a dedicated discussion through the end of Section 5.

This new paragraph explicitly highlights the following:

- Although turbulence distortion effects are already evident within the inflow turbulence noise bandwidth considered in this study, these effects tend to become more pronounced toward higher frequencies.

- When TE noise is included in predictions, its contribution to the noise generation may partially mask these effects on noise, particularly at higher frequencies.

- Thus, the impact of turbulence distortion may not be directly apparent in overall noise metrics unless noise components are examined separately.

A brief clarifying sentence has also been added to the Conclusions to guide interpretation of the results.